# GARLIC: LLM-Guided Dynamic Progress Control with Hierarchical Weighted Graph for Long Document QA

## Abstract

In the past, Retrieval-Augmented Generation (RAG) methods split text into chunks to enable language models to handle long documents. Recent tree-based RAG methods are able to retrieve detailed information while preserving global context. However, with the advent of more powerful LLMs, such as Llama 3.1, which offer better comprehension and support for longer inputs, we found that even recent tree-based RAG methods perform worse than directly feeding the entire document into Llama 3.1, although RAG methods still hold an advantage in reducing computational costs. In this paper, we propose a new retrieval method, called LLM-Guided Dynamic Progress Control with Hierarchical Weighted Graph (GARLIC), which outperforms previous state-of-the-art baselines, including Llama 3.1, while retaining the computational efficiency of RAG methods. Our method introduces several improvements: (1) Rather than using a tree structure, we construct a Hierarchical Weighted Directed Acyclic Graph with many-to-many summarization, where the graph edges are derived from attention mechanisms, and each node focuses on a single event or very few events. (2) We introduce a novel retrieval method that leverages the attention weights of LLMs rather than dense embedding similarity. Our method allows for searching the graph along multiple paths and can terminate at any depth. (3) We use the LLM to control the retrieval process, enabling it to dynamically adjust the amount and depth of information retrieved for different queries. Experimental results show that our method outperforms previous state-of-the-art baselines, including Llama 3.1, on two single-document and two multi-document QA datasets, while maintaining similar computational complexity to traditional RAG methods. [1]

## 1 Introduction

Retrieval-Augmented Generation (RAG) methods (Robertson et al., 1995; Robertson & Zaragoza, 2009; Reimers & Gurevych, 2019; Karpukhin et al., 2020; Khattab & Zaharia, 2020; Tay et al., 2022; Santhanam et al., 2022; Lin et al., 2023) have been a popular approach for handling QA tasks. Longer documents were segmented into chunks, and the most relevant chunks were retrieved and fed into a language model to generate answers. With the advent of Large Language Models (LLMs) (Touvron et al., 2023a;b), tree-based approaches such as RAPTOR (Sarthi et al., 2024) and MeMWalker (Chen et al., 2023a) have emerged. These models utilize LLMs to iteratively summarize the text, constructing tree-based summaries. By integrating information from different parts of the text, these methods facilitate the retrieval of both granular and high-level information, thereby improving performance through a balance of detailed understanding and global context, while managing longer documents effectively. However, as LLMs evolved, their capacity expanded, and models such as Llama 3.1 (Dubey et al., 2024) started supporting inputs of up to 128K tokens with enhanced comprehension capabilities. Studies like LongBench (Bai et al., 2024b) have demonstrated that the performance of RAG methods is often inferior to feeding the full document into LLMs directly (Zhang et al., 2023; Nair et al., 2023; Newman et al., 2023). Our experimental results also find that both RAPTOR and MeMWalker perform less well when compared to directly inputting the text into Llama 3.1. Nevertheless, retrieval methods remain beneficial in reducing input lengths and managing computational costs.

---

[1]The source code will be released upon paper acceptance.

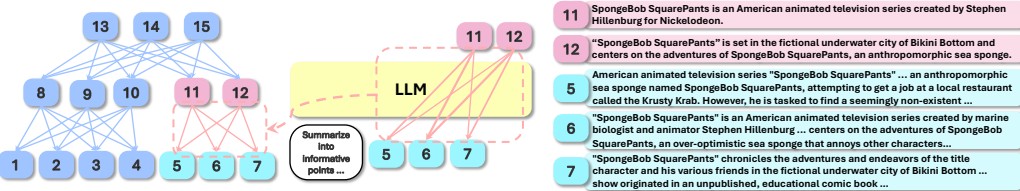

Figure 1: Overview of the Hierarchical Weighted Directed Acyclic Graph for summarization. Each node contains an Information Point (IP) and has multiple parent and child nodes, i.e., multiple successors and predecessors. Each time, the LLM is fed multiple nodes and prompted to generate multiple IPs. The weights of edges between nodes are computed based on the attention weights from LLM summarization. Some example IPs and chunks are shown on the right. For brevity, some long text is omitted.

In this paper, we introduce a new method, **LLM-Guided Dynamic Progress Control with Hierarchical Weighted Graph (`GARLIC`)**, which outperforms RAG baselines and even Llama 3.1 while preserving the benefits of retrieval with lower computational costs. Our method consists of two stages: *Summary Graph Construction* and *Dynamic Graph Search*. In the *Summary Graph Construction* stage, we prompt an LLM to generate multiple bullet-point sentences, termed **Information Points (IPs)**. Each IP typically focuses on a single or very few events. IPs are initially generated from text segments and then fed back into an LLM to recursively generate higher-level IPs. This process forms a many-to-many graph in which each node can have multiple lower-level child nodes and higher-level parent nodes, i.e., multiple successors and predecessors. During summarization, the LLM attention weights between the generated IPs and the input IPs are extracted to capture their relationships, as shown in Figure 1. The LLM consolidates and summarizes the same event from different sources, enabling the model to efficiently identify specific events rather than scanning through entire summary texts. The attention mechanism records which parts of the text contribute to an IP. The resulting structure is a *Hierarchical Weighted Directed Acyclic Graph* (HWDAG), where the extracted attention serves as the edge weights of the graph.

In the *Dynamic Graph Search* stage, we retrieve nodes from the HWDAG and feed them into the LLM to predict answers. We introduce a method called *LLM-guided Dynamic Progress Control*, which dynamically determines when to stop the search. Starting with top-level nodes, one node is selected per iteration. The LLM evaluates if the retrieved nodes contain sufficient information to answer a given query. The search continues until the LLM signals that enough information has been gathered. The prompt asks: "*Can this question be answered by the following documents?*" followed by the query and the text of the retrieved node. Each time a new node is retrieved, the previous inputs are KV cached, allowing new documents to be appended without reprocessing the entire input. Therefore, although we use LLM to decide when to stop the retrieval process, the method does not introduce additional computational overhead. This approach effectively resolves the prior challenge of determining how many chunks to search.

During the search, the attention between the retrieved nodes and the query is extracted, enabling it to assess relevance based on the LLM's knowledge. We combine this attention with the attention from the *Dynamic Graph Search* stage to guide the search and retrieve the next node. We employ Greedy Best-First Search (GBFS), a variant of Best-First Search (BFS), to traverse the graph. In contrast, RAPTOR (Sarthi et al., 2024) and MeMWalker (Chen et al., 2023a) utilize search methods that are more similar to Depth-First Search (DFS). We treat the graph as an adjacency matrix, considering all nodes connected to the retrieved node during the search, enabling the exploration of multiple paths. On the contrary, previous methods, as shown in Figure 2b, perform a search by following a single path from the top node to the bottom node since the number of nodes to retrieve was uncertain. Our method, however, is more flexible, as shown in Figure 2c, allowing for multiple paths and enabling the search to terminate at any level. Therefore, our method can handle queries that require varying amounts of information and information spread across different parts of the graph more effectively.

The combination of HWDAG and attention-based search enables us to introduce an alternative retrieval approach that solely relies on attention weights. GBFS integrates well with the HWDAG's structure, which consists of numerous smaller nodes, allowing the flexible retrieval of multiple IPs in any quantity and order. GBFS equipped with Dynamic Progress Control empowers the LLM to decide when to stop, making the retrieval process adaptable to different queries and allowing for

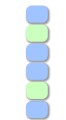 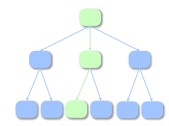 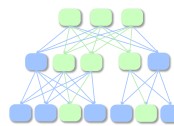

(a) Chunk-based retrieval. (b) Tree-based retrieval. (c) Our HWDAG-based retrieval.

Figure 2: Comparison of three retrieval methods. Nodes shaded in green are retrieved nodes. **(a) Chunk-based retrieval**. **(b) Tree-based retrieval**, starting from the top node and selecting a child node at each level until reaching the bottom node. **(c) Retrieval based on a HWDAG**. The node search is flexible, allowing multiple paths from the top level, and the search can stop at any level.

termination at various points along the search paths and at different levels of detail. In experiments, GARLIC outperforms other baselines, including Llama 3.1, without incurring additional inference computational costs. In summary, the contributions of this paper are as follows:

- We propose a novel Dynamic Progress Control mechanism using the LLM to control the retrieval process while employing KV caching to avoid additional time complexity.
- We propose a novel Attention-based Retrieval paradigm based exclusively on LLM attention weights using a Hierarchical Weighted Directed Acyclic Graph with many-to-many summarization. Each node represents an Information Point (IP) focused on a single or very few events, and the graph edges are derived from LLM attention during summarization.
- Our method surpasses previous state-of-the-art baselines, including Llama 3.1, while maintaining the computational efficiency of retrieval methods.

## 2 RELATED WORK

**Retrieval** Traditional retrieval techniques, such as TF-IDF (Jones, 1972) and BM25 (Robertson et al., 1995; Robertson & Zaragoza, 2009), retrieve information based on word terms. Subsequently, deep learning–based retrieval methods quickly became popular. REALM (Guu et al., 2020) augments the language model pre-training with a latent knowledge retriever using masked language modeling. DPR (Dense Passage Retrieval) (Karpukhin et al., 2020) encodes queries and documents as dense embeddings, with similarity computed between them. ColBERT (Khattab & Zaharia, 2020; Santhanam et al., 2022) produces multi-vector representations at the token level. JPR (Joint Passage Retrieval) (Min et al., 2021) is a joint passage retrieval model with an autoregressive reranker that selects a sequence of passages. DHR (Dense Hierarchical Retrieval) (Liu et al., 2021) leverages both macroscopic document-level semantics and microscopic passage-level semantics. Fusion-in-Decoder (Izacard & Grave, 2021) employs both DPR and BM25 in a knowledge distillation manner, which does not require annotated query-document pairs. CPT-text (Neelakantan et al., 2022) utilizes contrastive pre-training on unsupervised data. NCI (Wang et al., 2022) directly generates relevant document identifiers for a given query. Atlas (Izacard et al., 2022) fine-tunes an encoder-decoder model with a retriever to address knowledge-intensive tasks with minimal training examples. RETRO (Borgeaud et al., 2022; Wang et al., 2023a) conditions on document chunks based on local similarity with preceding tokens. HHR (Hybrid Hierarchical Retrieval) (Arivazhagan et al., 2023) combines sparse and dense retrieval methods across both document and passage retrieval stages. SimLM (Wang et al., 2023b) proposes a new loss function to reduce the mismatch between pre-training and fine-tuning input distributions. Dragon (Lin et al., 2023) uses contrastive learning and data augmentation to train a model, achieving state-of-the-art retrieval performance among eight baselines. Additionally, with the rise of LLMs, some research has explored the use of LLMs as retrievers. GENREAD (Yu et al., 2023) prompts LLMs to generate contextual documents based on a given query. RECITE (Sun et al., 2023) retrieves relevant passages from the LLM's internal memory via sampling. KGP (Knowledge Graph Prompting) (Wang et al., 2023c) builds a knowledge graph from multiple documents, with the LLM navigating. Recently, MeMWalker (Chen et al., 2023a) constructs tree-based summaries and uses LLMs to navigate through the tree. RAP-TOR (Sarthi et al., 2024) also creates tree-based summaries with clustering and uses embedding similarities to select the most relevant nodes at each level for retrieval. However, our approach differs from these methods. We construct a summary graph with IPs and employ attention mechanisms and GBFS for retrieval along any path, whereas MeMWalker and RAPTOR follow a single path from the top level to the bottom. Additionally, our method uses the LLM to dynamically determine when to stop the search, whereas MeMWalker also uses LLM to navigate but incurs significantly higher computational costs.

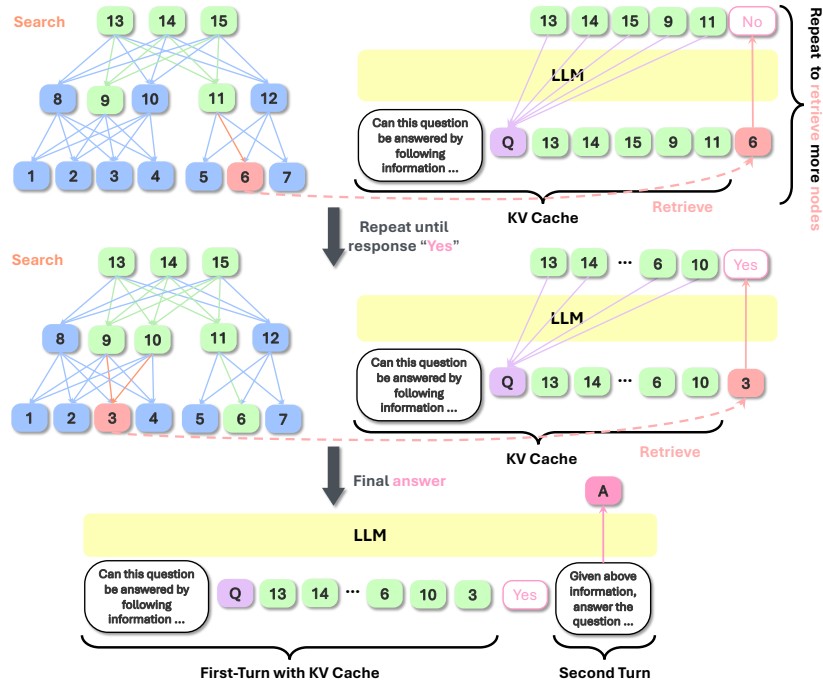

Figure 3: Overview of **Dynamic Graph Search**. Each time, a node is retrieved by Greedy Best-First Search using attention weights. The visited nodes are fed into the LLM, prompting the LLM to determine if sufficient nodes have been gathered to answer the query. This process incurs no additional computational cost due to KV caching. The search continues until the LLM signals that enough relevant nodes are retrieved, at which point the final answer is generated. The process adjusts dynamically based on the query, retrieving nodes flexibly across multiple graph paths and depths.

**Long-Context Language Models**   Recent long-context language models have focused on overcoming the limitations of context window size, primarily through positional interpolation to extend long-context capabilities by training on full-length texts. Chen et al. (2023b) used positional interpolation on RoPE (Rotary Position Embedding) (Su et al., 2023) to extend context length. Ding et al. (2024) proposed LongRoPE which performs direct extrapolation by rescaling RoPE with varied interpolation across RoPE dimensions at different token positions. Peng et al. (2024) and Fu et al. (2024) fine-tuned models on longer inputs and extended RoPE for longer contexts. LongLoRA (Chen et al., 2024) shifts sparse attention on LoRA (Hu et al., 2022) to extend model capacity for longer inputs. LongAlign (Bai et al., 2024a) constructs a long-context dataset, adopting packing and sorted batching strategies. PoSE (Zhu et al., 2024) manipulates position indices by skipping bias terms in each chunk. SkipAlign (Wu et al., 2024) synthesizes long-range dependencies from the aspect of position indices. Liu et al. (2024) showed that performance can degrade significantly when the position of relevant information is altered. Infini-Transformer (Munkhdalai et al., 2024) handles infinitely long inputs using compressive memory, masked local attention, and long-term attention mechanisms. Our method is complementary to these approaches. The LLMs used in these methods could serve as the base model in our approach to further reduce computational demands. Our focus is on utilizing LLMs effectively rather than improving the LLMs, and our method is compatible with these long-context LLMs.

## 3   METHODOLOGY

Our method consists of two main steps: *Summary Graph Construction* and *Dynamic Graph Search*. During *Summary Graph Construction*, as depicted in Figure 1, we iteratively construct an HWDAG from the documents, where the nodes represent IPs. In *Dynamic Graph Search*, as illustrated in Figure 3, given a query, we dynamically retrieve the IPs from the constructed HWDAG by performing a search guided by an LLM. Once enough nodes are retrieved, the LLM generates the final answer based on them. *Summary Graph Construction* is independent of the query, while *Dynamic Graph Search* is query-dependent.

## 3.1 SUMMARY GRAPH CONSTRUCTION

This step generates a HWDAG, defined as $\mathcal{G} = (\mathcal{V}, \mathcal{E})$, where $\mathcal{V}$ represents the collection of nodes, and $\mathcal{E}$ represents the collection of edges. Each node $v_i^l \in \mathcal{V}$ contains the text and is in level $l$. The subset of nodes at level $l$ is denoted by $\mathcal{V}_l \subset \mathcal{V}$. Each edge $e_{i,j} \in \mathcal{E}$ represents a scalar value indicating the relatedness between node $v_i$ and node $v_j$. Since $\mathcal{G}$ is a directed graph, $e_{i,j} \neq e_{j,i}$. By default, if there is no relationship between nodes $v_i$ and $v_j$, then $e_{i,j}, e_{j,i} = 0$.

The text content is initially split into chunks of 300 tokens, which serve as the first-level nodes $\mathcal{V}_1$. For each level, we iteratively summarize nodes from $\mathcal{V}_l$ by batching them and feeding them into the LLM to obtain the higher-level nodes $\mathcal{V}_{l+1}$, as illustrated in Figure 1. Specifically, we select nodes for each batch by sequentially choosing them until the total text content length exceeds a threshold $n_s$. This threshold $n_s$ includes both the prompt and the generated summary. These nodes are then input into the LLM. The prompt instructs the LLM to generate summaries in bullet-point format, with each point referred to as an **Information Point (IP)**. The LLM prompt used for summarization is shown in Table 4 in Appendix A.1, and a graph example is shown in Appendix D.

In previous retrieval methods, whether using chunks or summaries, multiple events are often contained in a single text chunk, requiring the retrieval of the entire chunk even if only one event is relevant (Karpukhin et al., 2020; Lin et al., 2023; Chen et al., 2023a; Sarthi et al., 2024). In contrast, each IP corresponding to each node in our method describes a single event and reduces the smallest retrievable unit from a chunk to an IP.

The attention from a higher-level node $v_i^{l+1}$ to a lower-level node $v_j^l$ is averaged across all tokens and layers to produce a scalar value $e_{i,j}$, which serves as the edge value from node $v_i$ to node $v_j$, where $\sum_j e_{i,j} = 1$. The value of $e_{i,j}$ represents how much information the higher-level node $v_i^{l+1}$ extracts from the lower-level node $v_j^l$. For example, in Figure 3, node $v_9$ directs to nodes $\{v_j\}_{j=1}^4$, with edge values $\{e_{9,j}\}_{j=1}^4$, and $\sum_{j=1}^4 e_{9,j} = 1$. Detailed computation can be found in Appendix B.1.

## 3.2 DYNAMIC GRAPH SEARCH

This section explains the process of retrieving nodes from the graph $\mathcal{G}$. An overview of dynamic graph search is illustrated in Figure 3. The process consists of two steps: *Dynamic Progress Control*, discussed in Section 3.2.1, and *Graph Search* in Section 3.2.2.

### 3.2.1 DYNAMIC PROGRESS CONTROL

This subsection describes the dynamic control of the search process. Initially, a visited set, denoted as $\mathcal{S} \subset \mathcal{V}$, which is initialized with the top-level nodes of $\mathcal{V}$, is fed into the LLM. We use a two-turn prompt system. In the first turn, the LLM is prompted to determine whether the current set of visited nodes $\mathcal{S}$ is sufficient to answer the query. The prompt asks the LLM, "*Can this question be answered by the following information?*", followed by the query and visited nodes $\mathcal{S}$. The complete prompt is shown in Table 5 in Appendix A.2, including an example from NarrativeQA (Kočiský et al., 2018). If the response is "No", the search continues, and the next node is retrieved. The details of the search process will be introduced in Section 3.2.2. The newly retrieved node is then appended to the end of the visited set $\mathcal{S}$, and the LLM is queried again. This process repeats until the LLM responds with "Yes". Throughout the search, all previous inputs, including the prompt, query, and visited nodes $\mathcal{S}$, are cached using KV caching, as illustrated in Figure 3. Additional details can be found in Appendix C. This ensures that no additional computational resources are required. Some LLMs may insert special tokens between the prompt and response, but these tokens are minimal, and the additional computation is negligible. Once the LLM responds with "Yes", the second turn of the prompt will ask the LLM to answer the query. At this point, the final answer is obtained.

This approach allows the LLM to dynamically determine the number of nodes needed for retrieval based on the query. Different queries may require varying amounts and types of content. For example, a query that requires only high-level information can be answered with just a few high-level nodes, whereas a query spanning multiple detailed aspects of the document may require the retrieval of both high- and low-level nodes. Previous methods typically had a fixed length of retrieved content, which could lead to either too much or too little information being retrieved for certain queries.

Similar to the concept of early stopping patience, we introduce a stop patience $p$. With this approach, the search stops after the LLM responds "Yes" $p$ times. We observed that increasing $p$ can even

further improve performance, though this creates a trade-off between performance gains and computational cost. Performance improvements tend to plateau when $p > 5$, while the computational cost continues to increase. See Section 4.3 for more details.

### 3.2.2 GRAPH SEARCH

This section describes the process of searching the graph $\mathcal{G} = (\mathcal{V}, \mathcal{E})$. Beyond the embedding similarity used in dense retrieval, as described in (Sarthi et al., 2024), we introduce another approach by leveraging the attention mechanisms of the LLM.

First, we define the adjacency matrix as $\boldsymbol{E} \in \mathbb{R}^{n_v \times n_v}$, where $\boldsymbol{E}_{i,j} = e_{i,j}$ and $n_v$ represents the number of nodes in $\mathcal{V}$. Each time, the visited nodes $\mathcal{S} \subset \mathcal{V}$ are fed into the LLM using the prompt shown in Table 5. The attention weight between the visited node $v_i$ and the query $q$ is extracted and averaged across tokens and layers, denoted as $a_i$, which represents the attention that node $v_i$ gives to the query $q$. Given that the query precedes the visited nodes in the prompt, nodes that appear later in the sequence may distribute some of their attention to earlier nodes. We observe that nodes positioned later in the sequence tend to have lower attention scores. To address this issue, we apply an empirical normalization to $a_i$ by multiplying of the corresponding position $v_i$, with $q$ occupying the first position. For example, in Figure 3, the extracted attention $a_{15}$ is multiplied by 3, as $a_{15}$ is in the third position. We found this adjustment yields good experimental results. After obtaining all attention values on all nodes, we construct a vector $\boldsymbol{a} \in \mathbb{R}^{n_v}$, where $\boldsymbol{a}_i = a_i$, and the remaining elements are set to 0, i.e., $\boldsymbol{a}_j = 0$ for $v_j \notin \mathcal{S}$.

Once the adjacency matrix $\boldsymbol{E} \in \mathbb{R}^{n_v \times n_v}$ and the query attention vector $\boldsymbol{a} \in \mathbb{R}^{n_v}$ are computed, the score vector $\boldsymbol{z} \in \mathbb{R}^{n_v}$ is calculated as follows: $\boldsymbol{z} = \boldsymbol{E}^T \boldsymbol{a}$, where $\boldsymbol{z}$ represents the score for each node, indicating how likely it is to be retrieved. Further details on the computation process can be found in Section B.2. The intuition behind this is that, if a node, i.e., an IP, is strongly correlated with the query, the details about this node will be more helpful for answering the query. $\boldsymbol{E}$ represents the relationships between nodes, while $\boldsymbol{a}$ highlights which of the currently visited nodes is more relevant. Through matrix multiplication, we can identify nodes that are related to the current query but are not yet part of the visited set $\mathcal{S}$. Nodes that are more closely related to the query $q$, will have their related successor nodes assigned higher scores. If a retrieved node is not relevant to the query, it will receive a low score in $\boldsymbol{z}$, thus preventing the search from continuing through that node. When sufficient relevant details are retrieved, Dynamic Progress Control will stop the search to avoid retrieving unnecessary details. We use $\boldsymbol{E}^T_{j,i}$ for the calculation of $\boldsymbol{z}$, which represents the attention received by $v_j$ from $v_i$, because the goal is to calculate candidate scores for each node, focusing on the nodes that are the focus of attention. If multiple nodes are highly related to $q$ and also connected to a successor, that successor node will receive a higher score, as it accumulates scores from multiple predecessors. For example, in Figure 3, $v_3$ will receive scores from both $v_9$ and $v_{10}$. We normalize $\boldsymbol{z}$ and the query-node embedding similarity so that the sum of their elements equals 1, and then we add the embedding similarity to $\boldsymbol{z}$ as the final score. The node $v \notin \mathcal{S}$ that is not yet in the visited set $\mathcal{S}$ and has the highest score is selected as the next node to retrieve.

## 4 EXPERIMENTS

### 4.1 SETUP

**Dataset**   We use two single-document QA and two multi-document QA datasets from LongBench (Bai et al., 2024b): **NarrativeQA** (Kočiský et al., 2018) is a single-doc QA dataset containing 1,567 stories, including full texts of books and movie transcripts. **Qasper** (Dasigi et al., 2021) is a single-doc QA dataset with 1,585 papers, designed to seek information present in the papers. **HotpotQA** (Yang et al., 2018) is a multi-doc QA dataset that contains 112,779 examples, focusing on multi-hop QA. **MuSiQue** (Trivedi et al., 2022) is a multi-doc QA dataset with 24,814 examples featuring 2-4 hop questions and six reasoning types. See Appendix E for more statistics.

**Metrics**   We use F1, ROUGE-L (Lin, 2004), and BLEU-4 (Papineni et al., 2002) as evaluation metrics. The final scores are computed using the evaluation source code from LongBench (Bai et al., 2024b) and Hugging Face Evaluate[2]. Additionally, we measure the average TFLOPs (Tera Floating Point Operations) during search and inference for each query.

---

[2]https://github.com/huggingface/evaluate

**Baseline** We employ both traditional and recent baselines: **BM25** (Robertson et al., 1995; Robertson & Zaragoza, 2009) is a bag-of-words based retrieval method that ranks documents based on the query terms appearing in them. **SBERT** (Reimers & Gurevych, 2019) is a dense retrieval method that employs dense embeddings obtained through the encoder model. **Dragon** (Lin et al., 2023) is a dense retrieval method that uses contrastive learning and data augmentation to train a model, achieving state-of-the-art retrieval performance among eight baselines. **LongLLMLingua** (Jiang et al., 2024) introduces question-aware compression based on LLMLingua (Jiang et al., 2023), a prompt compression method. **MeMWalker** (Chen et al., 2023a) processes context into a tree of summary nodes and navigates the tree to search for relevant information guided by the LLM, to handle long-text QA tasks within a limited input window. **RAPTOR** (Sarthi et al., 2024) constructs a tree by recursively embedding, clustering, and summarizing chunks of text for retrieval. RAPTOR has two variants: "tree traversal" (**RAPTOR-TT**) and "collapsed tree" (**RAPTOR-CT**). **Llama3.1-8B** (Dubey et al., 2024) is the 8B version of Llama3.1, which expands the context window to 128K tokens, allowing documents from the four datasets, except for some from NarrativeQA, to be directly fed into the model.

For all baselines and GARLIC, we use Llama3.1-8B (Dubey et al., 2024) as the LLM for both summarization and inference. For MeMWalker, since the source code was not released, we implemented it according to the paper using Llama3.1. For LongLLMLingua, which was originally proposed to use a smaller model to compress prompts for GPT-3.5, we used the published model Phi-2-2.7B, as provided by LongLLMLingua, to compress the text before inputting it into Llama3.1-8B. For other models, we ran their source code using Llama3.1-8B. For all inferences, we did not use Chain-of-Thought (CoT). We use SBERT (Reimers & Gurevych, 2019) as the retrieval model in GARLIC. In the experiments, we set the stop patience $p = 1$ by default and the length threshold $n_s$ to 8K. Across all datasets and steps of our method, including graph construction and search, the input window is capped by $n_s$ and can be processed using an NVIDIA A100 80G GPU. A summary graph example is illustrated in Appendix D.

## 4.2 MAIN RESULTS

The main results are shown in Table 1. Here, TFLOPs refers to the query-dependent inference. For MeMWalker, RAPTOR, and GARLIC, summarization TFLOPs are not included as summarization is query-independent. For BM25, SBERT, and Dragon, in addition to top-5, we also add a top-$X$ set to match the TFLOPs of GARLIC for a fair comparison. Specifically, we used Top-7, Top-14, Top-4, and Top-7 for NarrativeQA, Qasper, HotpotQA, and MuSiQue, respectively. A similar *top-X* applies to RAPTOR-CT, with Top-20, Top-42, Top-12, and Top-22, respectively. It is worth noting that our Dynamic Progress Control can determine the appropriate number of chunks or nodes to retrieve in a single pass, whereas these methods require extensive hyperparameter searches to find the optimal number. For Llama3.1-8B on NarrativeQA, we used 8 A100 80G GPUs, with CPU offloading, to handle the dataset. However, we could only process an input window of 100K tokens with these resources, resulting in 22.3% of documents being truncated. For LongLLMLingua, since the compression is query-dependent, we included the compression TFLOPs.

The performance of BM25, SBERT, and Dragon was relatively similar, with Dragon showing an advantage on NarrativeQA. Comparing the top-5 and top-$X$ results, we found that for Llama3.1-8B, retrieving more chunks generally leads to better results. LongLLMLingua achieves better results than Llama3.1-8B on HotpotQA, possibly because it reorders documents to place the most relevant content upfront, mitigating the lost-in-the-middle effect (Liu et al., 2024). However, for other datasets, the deletion of sentences and tokens in LongLLMLingua negatively impacts its performance. Given the small size difference between Phi-2 and Llama3.1-8B, the compression TFLOPs take up a significant portion of the overall computation. This may not reflect the intended use cases of LongLLMLingua, making efficiency comparisons challenging. MeMWalker did not perform well and lagged behind traditional retrieval methods. MeMWalker was initially designed to overcome the input length limitations for LLMs, but it struggles to navigate large trees effectively. It requires the LLM to generate correct responses and formats at every node, and when the tree becomes too large, navigation is prone to failure. This likely contributed to its poor performance on NarrativeQA. Additionally, its high computational complexity arises from the need to invoke the LLM at every node. RAPTOR outperformed other retrieval methods at lower TFLOPs. Its summarization is able to extract key information from the document, which improves performance and reduces TFLOPs. RAPTOR-TT is constrained by its fixed retrieval path, which sequentially retrieves nodes from the top level to the bottom. While RAPTOR-CT top-$X$ achieves higher per-

Table 1: F1 (%), ROUGE-L (%), BLEU-4 (%), and TFLOPs of baselines and GARLIC on NarrativeQA, Qasper, HotpotQA, and MuSiQue. TFLOPs are calculated during query-dependent inference. "Ratio" represents the ratio of the baselines' TFLOPs to GARLIC's TFLOPs. *Top-X* of BM25, SBERT, and Dragon denotes Top-7, Top-14, Top-4, and Top-7 for NarrativeQA, Qasper, HotpotQA, and MuSiQue, respectively, where the numbers are selected to ensure the baselines have similar TFLOPs to GARLIC for a performance comparison at the same TFLOPs. A similar *top-X* applies to RAPTOR-CT, with Top-20, Top-42, Top-12, and Top-22, respectively.

| Method | NarrativeQA | | | | | Qasper | | | | |
|---|---|---|---|---|---|---|---|---|---|---|
| | F1 | ROUGE-L | BLEU-4 | TFLOPs | Ratio | F1 | ROUGE-L | BLEU-4 | TFLOPs | Ratio |
| **BM25** top-5 | 52.7 | 51.8 | 13.4 | 26.7 | 0.86x | 41.0 | 39.6 | 21.0 | 26.3 | 0.39x |
| **SBERT** top-5 | 36.5 | 35.8 | 6.6 | 26.8 | 0.86x | 44.4 | 42.4 | 23.9 | 26.0 | 0.39x |
| **Dragon** top-5 | 53.8 | 52.9 | 13.6 | 26.9 | 0.87x | 43.0 | 41.4 | 22.8 | 24.5 | 0.36x |
| **RAPTOR-TT** | 40.6 | 39.8 | 7.8 | 20.3 | 0.65x | 42.1 | 40.1 | 17.2 | 17.7 | 0.26x |
| **RAPTOR-CT** | 48.6 | 47.8 | 11.8 | 17.9 | 0.58x | 44.6 | 42.7 | 19.5 | 16.6 | 0.25x |
| **LongLLMLingua** | 50.5 | 49.5 | 10.1 | 1789.4 | 57.72x | 43.2 | 43.0 | 21.0 | 159.7 | 2.39x |
| **MeMWalker** | 11.2 | 9.8 | 2.6 | 353.8 | 11.41x | 39.0 | 36.8 | 17.4 | 123.9 | 1.85x |
| **BM25** top-$X$ | 53.7 | 52.9 | 14.0 | 37.5 | 1.21x | 47.0 | 45.1 | 22.8 | 69.3 | 1.04x |
| **SBERT** top-$X$ | 39.5 | 38.8 | 7.3 | 37.5 | 1.21x | 46.6 | 44.5 | 23.3 | 68.9 | 1.03x |
| **Dragon** top-$X$ | 55.1 | 54.2 | 13.6 | 37.5 | 1.21x | 46.9 | 44.8 | 22.1 | 67.0 | 1.00x |
| **RAPTOR-CT** top-$X$ | 52.0 | 51.2 | 11.8 | 35.1 | 1.13x | 46.9 | 44.7 | 20.8 | 67.3 | 1.01x |
| **Llama3.1-8B** | 53.7 | 52.6 | 10.4 | 3361.9 | 108.45x | 49.4 | 47.6 | 26.9 | 92.5 | 1.38x |
| **GARLIC** | 61.1 | 60.2 | 18.6 | 31.0 | 1.00x | 49.7 | 47.9 | 27.0 | 66.9 | 1.00x |

| Method | HotpotQA | | | | | MuSiQue | | | | |
|---|---|---|---|---|---|---|---|---|---|---|
| | F1 | ROUGE-L | BLEU-4 | TFLOPs | Ratio | F1 | ROUGE-L | BLEU-4 | TFLOPs | Ratio |
| **BM25** top-5 | 40.8 | 40.9 | 7.7 | 22.9 | 1.43x | 28.7 | 28.7 | 5.1 | 26.3 | 0.85x |
| **SBERT** top-5 | 40.9 | 40.8 | 8.0 | 22.6 | 1.41x | 30.7 | 30.8 | 6.3 | 26.1 | 0.84x |
| **Dragon** top-5 | 39.7 | 39.6 | 6.9 | 23.3 | 1.46x | 28.5 | 28.4 | 5.4 | 28.1 | 0.91x |
| **RAPTOR-TT** | 38.6 | 38.5 | 6.7 | 8.4 | 0.53x | 29.3 | 29.3 | 4.7 | 12.6 | 0.41x |
| **RAPTOR-CT** | 40.9 | 40.4 | 7.2 | 15.3 | 0.96x | 31.5 | 31.5 | 5.5 | 16.1 | 0.52x |
| **LongLLMLingua** | 43.4 | 43.5 | 8.1 | 43.6 | 2.73x | 34.5 | 34.4 | 5.6 | 78.9 | 2.55x |
| **MeMWalker** | 39.7 | 38.9 | 13.9 | 93.4 | 5.84x | 24.0 | 23.5 | 9.9 | 175.7 | 5.69x |
| **BM25** top-$X$ | 40.7 | 40.8 | 7.7 | 20.0 | 1.25x | 31.8 | 31.7 | 5.6 | 35.6 | 1.15x |
| **SBERT** top-$X$ | 40.8 | 40.7 | 7.5 | 19.6 | 1.23x | 32.5 | 32.5 | 6.4 | 35.6 | 1.15x |
| **Dragon** top-$X$ | 39.2 | 39.1 | 6.7 | 20.6 | 1.29x | 30.2 | 30.1 | 6.0 | 38.0 | 1.23x |
| **RAPTOR-CT** top-$X$ | 40.7 | 40.7 | 7.2 | 17.9 | 1.12x | 35.4 | 35.2 | 7.2 | 32.2 | 1.04x |
| **Llama3.1-8B** | 41.3 | 41.2 | 6.3 | 23.7 | 1.48x | 35.8 | 35.7 | 5.6 | 40.6 | 1.31x |
| **GARLIC** | 43.5 | 43.5 | 7.2 | 16.0 | 1.00x | 36.9 | 36.8 | 5.7 | 30.9 | 1.00x |

formance by using more nodes, it still underperforms compared to our method at similar TFLOPs, demonstrating that its tree-based summarization is not as efficient as ours. Llama3.1-8B's excelled on most datasets except NarrativeQA, where input truncation likely affected its performance. For particularly long inputs in NarrativeQA, Llama3.1-8B reached 3361.9 TFLOPs due to an average token length of 794,457, suggesting that directly feeding very long texts into an LLM may not be the optimal choice.

Our method outperformed all baselines and Llama3.1-8B across all four datasets. Even with the same TFLOPs, our results were better than those of BM25, SBERT, and Dragon. Compared to LongLLMLingua, our method achieved better performance with lower TFLOPs, although the differing application scenarios limit direct comparison. Compared to MeMWalker and Llama3.1-8B, our method achieved higher performance at a lower computational cost. Similarly, our method outperformed RAPTOR while omitting the clustering step, as our retrieval process stops only after gathering sufficient nodes. Overall, our method demonstrates both performance advantages and low computational complexity. Its superiority over Llama3.1-8B can be attributed to the effective summarization of document key points through IPs, and the utilization of attention to enhance search effectiveness, combined with Dynamic Progress Control to ensure adequate information collection.

## 4.3 DYNAMIC SEARCH STOP STUDY

Following the concept of early stop patience, the stop patience $p$ in this paper refers to the number of times the LLM responds "Yes" before the search stops, as introduced in Section 3.2.1. In Table 1, we set the stop patience $p = 1$. In this section, we investigate how the stop patience $p$ influences both performance and efficiency.

As shown in Figure 4, increasing $p$ can further improve performance beyond the results in Table 1, but at the cost of increased computational resources. It can be observed that with our method and

Llama3.1, retrieving more nodes allows the model to provide more accurate answers. When $p < 5$, the F1 score increases rapidly, indicating that, on average, there are still up to 5 nodes containing useful information that have not yet been retrieved. However, when $p > 5$, the improvement in the F1 score slows, and the TFLOPs curve begins to exceed the F1 increase, suggesting that the additional retrieved nodes do not contribute significantly to the model's performance, thus reducing the cost-effectiveness of further computation. The slowdown in performance improvement as $p$ increases also validates the LLM's ability to stop the search, as few additional nodes are beneficial once the search stops. By adjusting $p$, GARLIC can balance between effectiveness and efficiency. Due to the presence of Dynamic Progress Control, there is a lower bound for this adjustment range, specifically when $p = 1$, and across all four datasets, performance increases more rapidly when $p < 5$. In contrast, previous methods required adjusting the number of chunks to retrieve based on different data distributions or even individual queries, resulting in higher hyper-parameter search costs.

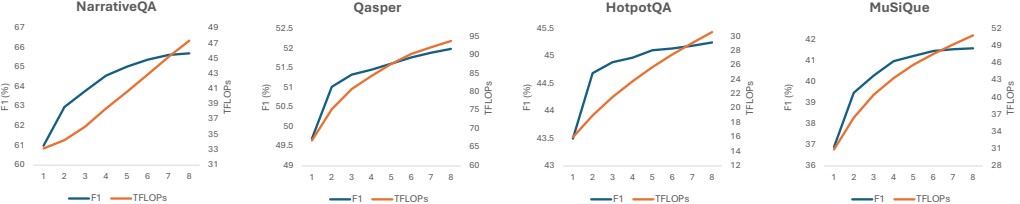

Figure 4: The F1 (%) and TFLOPs of GARLIC on NarrativeQA, Qasper, HotpotQA, and MuSiQue with different stop patience values $p$. The horizontal axis represents stop patience $p$. The left vertical axis shows the F1 (%) corresponding to the blue line, and the right vertical axis shows TFLOPs corresponding to the red line. As $p$ increases from 1 to 8, both F1 and TFLOPs increase, but the increase in F1 slows when $p > 5$, while TFLOPs continue to rise.

## 4.4 Ablation Study

In this section, we study how each component contributes to performance, as shown in Table 2, and describe them below.

Table 2: Ablation study of the four components of GARLIC with F1 (%), ROUGE-L (%), and BLEU-4 (%) on NarrativeQA, Qasper, HotpotQA, and MuSiQue.

| Method | NarrativeQA | | | Qasper | | | HotpotQA | | | MuSiQue | | |
|---|---|---|---|---|---|---|---|---|---|---|---|---|
| | F1 | ROUGE-L | BLEU-4 | F1 | ROUGE-L | BLEU-4 | F1 | ROUGE-L | BLEU-4 | F1 | ROUGE-L | BLEU-4 |
| GARLIC | 61.1 | 60.2 | 18.6 | 49.7 | 47.9 | 27.0 | 43.5 | 43.4 | 7.2 | 36.9 | 36.9 | 5.7 |
| w/o Graph-based Summary | 51.4 | 50.8 | 9.2 | 47.3 | 45.5 | 23.8 | 40.9 | 40.7 | 6.4 | 32.3 | 32.3 | 4.1 |
| w/o Dynamic Progress Control | 53.5 | 52.9 | 16.0 | 37.7 | 36.4 | 20.8 | 39.4 | 39.3 | 5.9 | 27.0 | 27.1 | 3.4 |
| w/o Attention Search | 53.0 | 52.4 | 18.3 | 46.9 | 45.5 | 25.3 | 41.9 | 41.7 | 6.8 | 33.1 | 32.9 | 5.1 |
| w/o Embedding Similarity Search | 59.5 | 58.7 | 17.6 | 48.0 | 46.2 | 23.5 | 42.9 | 42.8 | 6.6 | 35.9 | 35.8 | 5.0 |

**w/o Graph-based Summary**: We drop the graph-based summary and instead follow a tree-based manner, as illustrated in Figure 2b, by iteratively summarizing nodes and conducting the search with it. All datasets show a decline in performance, especially for NarrativeQA. For more complex and longer inputs, IPs help organize information more effectively.

**w/o Dynamic Progress Control**: We measured the average number of nodes used across the four datasets with GARLIC, which were 48, 36, 17, and 30 for NarrativeQA, Qasper, HotpotQA, and MuSiQue, respectively. Instead of using the LLM to dynamically decide when to stop the search, we employ a fixed number of nodes based on these averages. The search stops after retrieving this predefined number of nodes. For documents with many top-level nodes, the initial number of visited nodes $\mathcal{S}$ is limited to the preset value divided by the number of levels, ensuring sufficient room to search. The most relevant top-level nodes are selected using embedding similarity, similar to RAPTOR (Sarthi et al., 2024). We retain the same search mechanism using attention and embedding similarity, so unselected top-level nodes can still be retrieved via embedding similarity. Without Dynamic Progress Control, performance dropped across all datasets, even though the average number of nodes retrieved remained unchanged. Some queries retrieve unnecessary information, while others still lack the required information, leading to decreased performance. GARLIC dynamically adjusts the amount of information retrieved for each query, ensuring a proper amount is retrieved.

**w/o Attention Search**: We remove the use of attention and rely solely on embedding similarity for node search, similar to RAPTOR (Sarthi et al., 2024). Performance dropped across all datasets, with the most significant drop occurring in NarrativeQA, indicating that attention scores are particularly effective in retrieving hierarchical information from long texts.

**w/o Embedding Similarity Search**: We exclude embedding similarity from the final score $z$, relying entirely on attention weights to compute retrieval scores. While performance decreased slightly, it was not a major drop. The score for NarrativeQA was not significantly affected, indicating that while embedding similarity is beneficial, attention-based search plays a more critical role.

## 4.5 EFFICIENCY ANALYSIS

Table 1 lists the TFLOPs for search and inference for each query. In this section, we provide a closer analysis of efficiency. Table 3 shows the graph construction TFLOPs per document as GARLIC *graph construction* and the graph search TFLOPs per query as GARLIC *graph search*. GARLIC *graph construction + graph search* shows the average TFLOPs per query when each document contains 2, 4, or 8 queries.

Table 3: TFLOPs for graph construction and graph search. GARLIC *graph construction + graph search* shows the average TFLOPs per query when each document contains 2, 4, or 8 queries.

| Method | NarrativeQA TFLOPs | Qasper TFLOPs | HotpotQA TFLOPs | MuSiQue TFLOPs |
|---|---|---|---|---|
| **Llama3.1-8B** | 3361.9 | 92.5 | 23.6 | 40.6 |
| **GARLIC** graph construction | 2042.8 | 136.6 | 40.2 | 66.7 |
| **GARLIC** graph search | 31.0 | 66.9 | 16.0 | 30.9 |
| **GARLIC** graph construction + graph search | | | | |
| 2 queries per document | 1052.4 | 135.2 | 36.1 | 64.3 |
| 4 queries per document | 541.7 | 101.1 | 26.1 | 47.6 |
| 8 queries per document | 286.4 | 84.0 | 21.0 | 39.2 |

If each document has only one query, the TFLOPs of our method exceed those of Llama3.1 on Qasper, HotpotQA, and MuSiQue. During graph construction, the entire document is processed by the GARLIC, along with the additional summary generation. However, Llama3.1 uses more TFLOPs on NarrativeQA than GARLIC as the complexity of the Transformer (Vaswani et al., 2017) increases quadratically with input length for very long inputs. Even though the total amount of text processed by our method is longer, Llama3.1 processes the entire input at once, whereas our method processes the document in chunks, resulting in lower TFLOPs. As the number of queries per document increases, the TFLOPs for graph construction are amortized, reducing the average TFLOPs per query. When each document has more than 8 queries, our method achieves lower average TFLOPs per query, even when accounting for the summary.

Additionally, GARLIC is able to use an input window of 8K, allowing it to run on an NVIDIA A100 80G GPU. In contrast, we had to use 8 A100 80G GPUs, even with CPU offloading, to run Llama3.1 on NarrativeQA with an input window of 100K tokens. This limits the practical application of Llama3.1 in real-world scenarios, whereas our method can easily run on a single GPU.

## 5 CONCLUSION AND FUTURE WORK

In this paper, we propose a new retrieval method, called GARLIC, which constructs a Hierarchical Weighted Directed Acyclic Graph with many-to-many summarization. Each node represents an Information Point focusing on a single or few events, with edges derived from summarization attention. We also introduce a novel retrieval method using LLM attention weights and LLM-controlled retrieval, with efficiency maintained through KV caching. Our Dynamic Progress Control can be easily adapted to other retrieval methods. Experiments show our method outperforms baselines while maintaining retrieval computational efficiency, and increasing stop patience can further improve performance. There are some areas that could be improved. As we placed the query at the beginning, we had to apply an empirical normalization, which might not be optimal. Additionally, the equal weighting of attention-based and embedding similarity scores could be refined. Further work could explore incorporating Chain-of-Thought (CoT) reasoning to further enhance performance.

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

## A    PROMPT

### A.1    SUMMARY GRAPH CONSTRUCTION PROMPT

In *Summary Graph Construction*, the prompt instructs the LLM to generate summaries in bullet-point format, with each point referred to as an Information Point (IP). The LLM prompt used is shown in Table 4, with an example from NarrativeQA (Kočiský et al., 2018). In the prompt, we use the term "summary point" to refer to the IPs in the response, distinguishing them from the IPs in the input for LLM. Each IP in our method describes a single or very few events, and during retrieval, each IP, corresponding to each node, reduces the smallest retrievable unit from a chunk to an IP. An IP usually consists of just one sentence, allowing our method to retrieve specific pieces of knowledge more efficiently.

Table 4: The prompt used for the LLM during *Summary Graph Construction*, with an example from NarrativeQA. The *italicized* text represents the example context, while the remaining text is the prompt instruction. To differentiate between the input and output IPs, the output IPs are named as "summary points" in the prompt to avoid confusion for LLM. Each summary point starts with an asterisk and typically consists of a single sentence.

| **Summary Graph Construction** |
|---|
| **Prompt:** |
| Summary the following information. Each segment is separated by a new line symbol. |
| ...... 
 *\* Mrs. Tabitha Twitchit expects "fine company" for tea and fetches the children before her friends arrive.* 
 *\* Tabitha dresses Moppet and Mittens in clean pinafores and tuckers, and Tom in "all sorts of elegant uncomfortable clothes" taken from a chest of drawers.* 
 *\* Tom is fat and bursts several buttons, but his mother sews them back on again.* 
 *\* Tabitha turns her kittens into the garden to keep them out of the way while she makes hot buttered toast for the party.* 
 ...... |
| Split your summary into different summary points according to the semantic information in these information points. It is not necessary to generate each summary point for each information point. Gather and organize information into summary points. In each summary point, try to avoid using pronouns like he/she/they and instead use full names. Generate in the format of: |
| \* summary point 
 \* summary point 
 \* summary point 
 ...... |
| Do not provide any explanation and start the summary directly. |
| **Response:** |
| *\* Mrs. Tabitha Twitchit expects fine company for tea and dresses Mittens, Tom Kitten, and Moppet in clean clothes.* 
 *\* Mrs. Tabitha Twitchit sends Mittens, Tom Kitten, and Moppet to the garden to keep them out of the way.* 
 ...... |

### A.2    DYNAMIC GRAPH SEARCH PROMPT

The two-turn prompt is shown in Table 5 in Appendix A.2, including an example from NarrativeQA (Kočiský et al., 2018). In the first turn, the LLM is prompted to determine whether the current set of visited nodes $\mathcal{S}$ is sufficient to answer the query. The prompt asks the LLM, "*Can this question be answered by the following information?*", followed by the query and visited nodes $\mathcal{S}$. If the response is "No", the search continues, and the next node is retrieved. This process repeats until the LLM responds with "Yes". Once the LLM responds with "Yes", the second turn of the prompt will ask the LLM to answer the query.

Table 5: The two-turn prompt used for *Dynamic Graph Search*, with an example from NarrativeQA. The *italicized* text represents the example context, while the remaining text is the prompt instruction. The first-turn prompt asks the LLM to determine if the current context is sufficient to answer the query. If the response is "No," more context will be added to the end of the prompt, and the LLM will be asked again, with all previous context KV cached to avoid additional computation. Once the LLM responds with "Yes," the second turn will prompt the LLM to provide the final answer directly.

---

**Dynamic Graph Search**

**First-Turn Prompt:**

Can this question be answered by the following information? Response "Yes" or "No" in one word. Do not provide any explanation.

Question:
*Where does the mother send her kittens to keep them out of the way while getting ready for the party?*

Information:
......
*\* Mrs. Tabitha Twitchit sends Mittens, Tom Kitten, and Moppet to the garden to keep them out of the way.*
*\* Tabitha turns her kittens into the garden to keep them out of the way while she makes hot buttered toast for the party.*
......

**First-Turn Response:**

Yes

**Second-Turn Prompt:**

Given the above information and question, answer the question as concisely as you can.

**Second-Turn Response:**

*The garden.*

---

# B   ATTENTION COMPUTATION

## B.1   AVERAGING AND NORMALIZATION

This section introduces the detailed computation of the final attention weight $e_{i,j} \in \mathcal{E}$ between a high-level node $v_i^{l+1}$ and a low-level node $v_j^l$, as introduced in Section 3.1. We define the lower-level node $v_j^l$ with tokens $\{x_k^j\}_{k=1}^{K_j}$ and the higher-level node $v_i^{l+1}$ with tokens $\{x_k^i\}_{k=1}^{K_i}$, where $K_j$ and $K_i$ denote the number of tokens in nodes $v_j^l$ and $v_i^{l+1}$, respectively.

First, we extract all attention weights from the LLM during summarization. These attention weights are averaged across attention heads and layers. In practice, we iteratively accumulate attention weights averaged over attention heads for each layer to reduce memory usage. This process yields the attention weight $e_{x^i,x^j}$ between tokens $x^i \in v_i^{l+1}$ and $x^j \in v_j^l$.

Next, we average attention weights at the token level within each node. For the lower-level node $v_j^l$, we average $\{e_{x^i,x_k^j}\}_{k=1}^{K_j}$ into $e_{x^i,v_j^l}$, which represents the attention weight from token $x^i \in v_i^{l+1}$ to the node $v_j^l$. Similarly, for the higher-level node $v_i^{l+1}$, we average $\{e_{x_k^i,v_j^l}\}_{k=1}^{K_i}$ into $e_{v_i^{l+1},v_j^l}$. For simplicity, we refer to $e_{v_i^{l+1},v_j^l}$ as $e_{i,j}$ in Section 3.1.

This process is repeated for all nodes. If no connection exists between two nodes, the edge weight is set to zero. Finally, the attention weights are normalized as $e_{i,j} = \frac{e_{i,j}}{\sum_j e_{i,j}}$, ensuring that $\sum_j e_{i,j} = 1$. This provides the final attention weights for the edges $\mathcal{E}$ in the graph $\mathcal{G}$. Similarly, the query-based attention $\boldsymbol{a}$ described in Section 3.2.2 is computed in this manner.

We classify attention into two types: syntactic attention, which reflects grammatical relationships, and semantic attention, which captures meaning connections, often between paragraphs. When

averaging token-level attentions, we only extract attentions between high-level and low-level nodes, omitting attentions within the same node. This process emphasizes long-distance semantic attention, which effectively reflects the semantic relationships between nodes.

## B.2 Retrieval Score Computation

In this section, we introduce the computation of $z = E^T a$ from a different perspective.

The final retrieval score $z$ is obtained as the product of the adjacency matrix $E^T$ and the query relation vector $a$. Specifically, for a node $v_i$, its predecessors are defined in the set $\mathcal{P}_i$. We identify the intersection of $\mathcal{P}_i$ and the set of visited nodes $\mathcal{S}$, denoting it as $\mathcal{R}_i = \mathcal{P}_i \cap \mathcal{S}$.

Given nodes $v_j \in \mathcal{R}_i$, retrieval score $z_i$ of a node $v_i \notin \mathcal{S}$ is computed as:

$$z_i = \sum_j e_{j,i} a_j, \tag{1}$$

where $e_{j,i}$ represents the edge weight from node $v_j$ to node $v_i$. Once this operation is performed for all nodes, the vector $z$ is fully computed.

This computation aligns with $z = E^T a$ described in Section 3.2.2. When the adjacency matrix $E$ is sparse or the set $\mathcal{S}$ is small, leveraging sparse matrix multiplication or directly using Eq. (1) can be more computationally efficient.

## C KV Caching in Dynamic Progress Control

This section introduces the usage of the KV cache described in Section 3.2.1. Given the last retrieved node $v_i$ in the visited set $\mathcal{S}$, let $\{x_k^i\}_{k=1}^{K_i}$ denote the tokens of node $v_i$, where $K_i$ represents the number of tokens in $v_i$.

The query and visited nodes are represented as $[q, v_1, \ldots, v_i, \ldots, v_S]$, with their corresponding tokens organized as $[x_1^q, \ldots, x_{K_q}^q, x_1^1, \ldots, x_{K_1}^1, \ldots, x_1^i, \ldots, x_{K_i}^i]$.

When a new node $v_j$ is retrieved, the query, key, and value states of the tokens $\{x_k^j\}_{k=1}^{K_j}$ in $v_j$ are computed using the self-attention mechanism of the LLM. Since $v_j$ also attend to the previously visited nodes, the stored KV cache containing the tokens $[x_1^q, \ldots, x_{K_q}^q, x_1^1, \ldots, x_{K_1}^1, \ldots, x_1^i, \ldots, x_{K_i}^i]$ is input into the LLM. At this point, the query states from $v_j$ and the key-value states from $q$, $\mathcal{S}$, and $v_j$ are available. The LLM performs self-attention over these query, key, and value states.

After processing, $v_j$ is added to the visited set $\mathcal{S}$, and the key-value states of its tokens $\{x_k^j\}_{k=1}^{K_j}$ are stored. These states are concatenated with the previous KV cache to form $[x_1^q, \ldots, x_{K_q}^q, x_1^1, \ldots, x_{K_1}^1, \ldots, x_1^i, \ldots, x_{K_i}^i, x_1^j, \ldots, x_{K_j}^j]$. This updated KV cache is then used for the next node retrieval. In this common scenario, KV caching is typically performed at the token level, where the LLM caches the key-value states of previous tokens when generating the next token. In contrast, our approach utilizes a node-level KV cache for retrieval.

Once the retrieval process is complete, the key-value states of all visited nodes in $\mathcal{S}$ are cached, and the LLM is prompted to answer the question using these cached states. The LLM follows standard decoding to generate each token one at a time, leveraging the KV cache of the previous tokens. The query, key, and value states for all nodes, the query, and the answers are computed only once throughout the retrieval process to avoid additional computation.

## D Summary Graph Example

In this section, we present an example of generated IPs with part of the first and second-level nodes in Figure 5 from HotpotQA (Yang et al., 2018). The nodes on the right are higher-level IPs generated from the low-level chunk nodes on the left. The connections in the middle represent attention weights. The higher the attention weight, the redder and thicker the line. Lines with attention weights less than 0.05 are omitted from the figure. It can be observed that IPs are generated from multiple chunks. For example, high-level nodes 10, 13, 14, 15, 16, 17, 18, and 19 are connected to multiple low-level nodes. In contrast, high-level nodes 11 and 20 mainly rely on single low-level

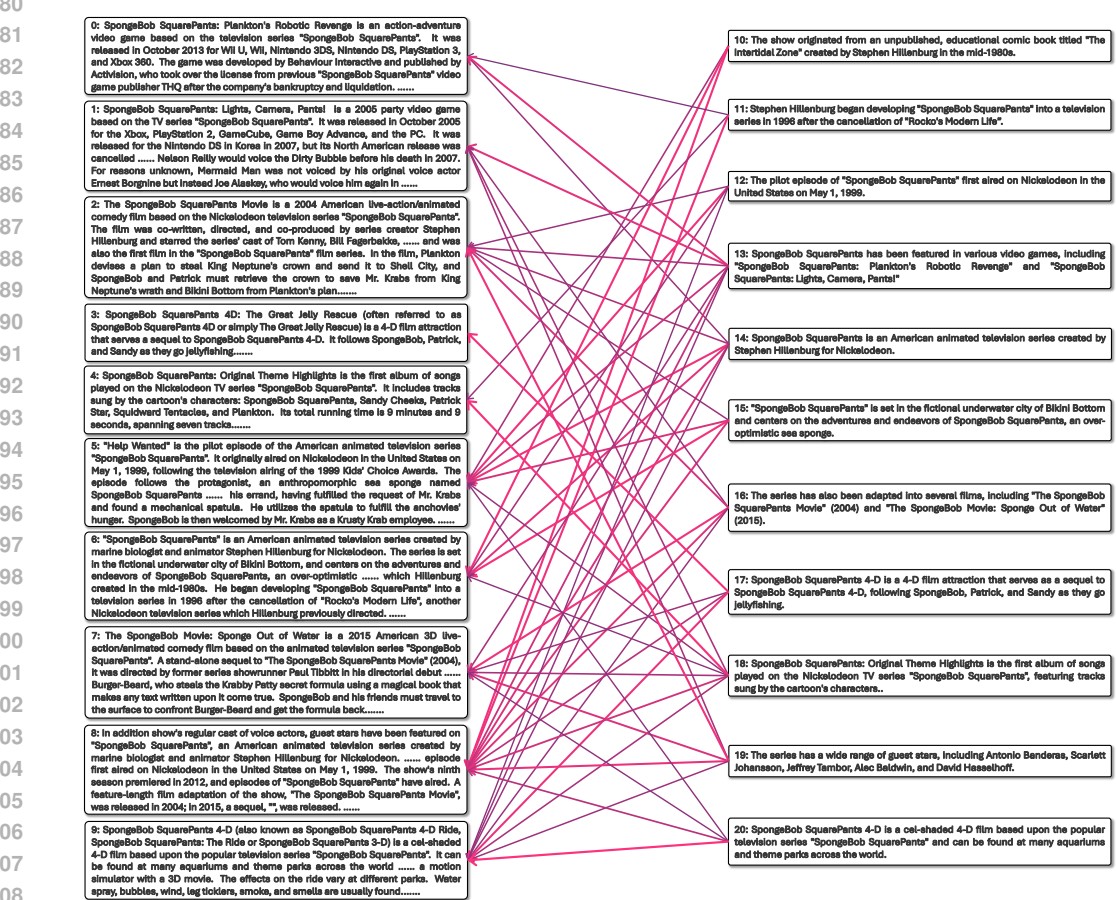

Figure 5: An example of generated IPs with part of the first and second-level nodes from HotpotQA. The nodes on the right are higher-level IPs generated from the low-level chunk nodes on the left. The connections in the middle represent attention weights. The higher the attention weight, the redder and thicker the line. Lines with attention weights less than 0.05 are omitted from the figure. For brevity, some long text in the nodes on the left is omitted.

nodes, with minimal connections to other low-level nodes. From the perspective of low-level nodes, nodes 2, 5, 6, 7, 8, and 9 are connected to many high-level nodes. Meanwhile, low-level node 3 is only attended to by node 17. Different nodes automatically adjust their relevance based on attention with other nodes. Some nodes connect to multiple nodes, while others connect to a single node. Our method uses IPs and attention to make the relationships between low-level nodes and higher-level nodes more explicit, allowing the model to understand how each piece of information is connected to different parts of the text through the graph.

# E   DATASETS STATISTICS

Table 6 shows the token length statistics of the datasets NarrativeQA (Kočiský et al., 2018), Qasper (Dasigi et al., 2021), HotpotQA (Yang et al., 2018), and MuSiQue (Trivedi et al., 2022). NarrativeQA is much longer than the other datasets, followed by Qasper. HotpotQA and MuSiQue are relatively shorter.

Table 6: Token length statistics for the NarrativeQA, Qasper, HotpotQA, and MuSiQue datasets, including the average, minimum, and maximum token length per document.

| Dataset | Average | Min | Max |
|---|---|---|---|
| **NarrativeQA** (Kočiský et al., 2018) | 79,457 | 5,077 | 467,867 |
| **Qasper** (Dasigi et al., 2021) | 4,866 | 918 | 29,408 |
| **HotpotQA** (Yang et al., 2018) | 1,318 | 70 | 3,575 |
| **MuSiQue** (Trivedi et al., 2022) | 2,267 | 909 | 4,432 |

