# OpenReview forum: "GARLIC: LLM-Guided Dynamic Progress Control with Hierarchical Weighted Graph for Long Document QA"
_ICLR.cc/2025/Conference — Submitted to ICLR 2025_

### Official Review · Reviewer_LAnf · 2024-10-29

**Soundness:** 2
**Presentation:** 2
**Contribution:** 3
**Rating:** 6
**Confidence:** 3

**Summary:**

The authors propose a method using LLMs to summarize long text into key information points, organizing these points into a hierarchical structure through recursive processing. This results in a directed acyclic graph where edge weights between nodes are generated based on attention mechanisms. For QA tasks, the authors apply a weighted BFS to retrieve relevant information from the graph. An LLM dynamically controls the retrieval process by determining if the information collected so far is sufficient to answer the question.

**Strengths:**

1. The system uses a large language model to recursively generate key information from long texts, constructing an information structure graph. It employs the model's attention mechanism to assign weights to edges, introducing a new structure for information representation.

2. The authors propose using a priority-based breadth-first search to retrieve information points within the graph. The search process is controlled by the large language model, allowing retrieval to stop as soon as relevant information is found. This approach, compared to depth-first search, offers greater flexibility while maintaining efficiency. Final experimental results outperform recent benchmarks, including Llama 3.1.

**Weaknesses:**

1. The paper does not explain how edge weights between nodes are calculated using the attention mechanism of the LLM, lacking crucial algorithmic details.

2. In Section 3.2.2, the paper discusses calculating the relevance between the query and information points using attention. How is this calculation performed, especially if it needs to be precomputed? This method may have limited scalability for new queries and new documents.

3. No code is released, making it difficult for others to use the proposed method or reproduce the result.

**Questions:**

see weakness

---

> ### Author Response · Authors · 2024-11-17
>
> We appreciate the reviewer's time and the thoughtful comments. The response is as follows:
>
> ---
> > **Q1**: The paper does not explain how edge weights between nodes are calculated using the attention mechanism of the LLM, lacking crucial algorithmic details.
>
> **A1**:
> Given input tokens such as $[a_1, a_2], [b_1, b_2]$ for nodes $a$ and $b$, and output tokens $[c_1, c_2], [d_1, d_2]$ for nodes $c$ and $d$, edge weights are calculated as follows:
>
> 1. The LLM generates many-to-many summaries from $[a_1, a_2], [b_1, b_2]$ to $[c_1, c_2], [d_1, d_2]$.
> 2. During processing, attention scores are recorded and averaged across all attention heads and layers. For example, attention weights $e_{c_1 \rightarrow b_1}$ across all heads and layers are averaged into a single attention weight $e_{c_1 \rightarrow b_1}$ for the summary.
> 3. Token-level attention weights are grouped and averaged by node. For example, attention weights $e_{c_1 \rightarrow b_1}$ and $e_{c_1 \rightarrow b_2}$ are averaged to yield a single weight $e_{c_1 \rightarrow b}$. This is repeated for all input nodes:  $e_{c_1 \rightarrow a}$, $e_{c_2 \rightarrow a}$, $e_{d_1 \rightarrow a}$, $e_{d_2 \rightarrow a}$, $e_{c_1 \rightarrow b}$, $e_{c_2 \rightarrow b}$, $e_{d_1 \rightarrow b}$, $e_{d_2 \rightarrow b}$.
> 4. Similarly, for output $d$, we average $e_{c_1 \rightarrow b}$ and $e_{c_2 \rightarrow b}$ into $e_{c \rightarrow b}$. This step is repeated for all output nodes: $e_{c \rightarrow a}$, $e_{d \rightarrow a}$, $e_{c \rightarrow b}$,  $e_{d \rightarrow b}$.
> 5. Finally,  output-level weights are normalized to sum to 1. For example, for output $d$, $e_{c \rightarrow a}$, and $e_{c \rightarrow b}$, are normalized to sum to 1, using $e_{c \rightarrow i} = \frac{e_{c \rightarrow i}}{\sum e_{c \rightarrow j}}$, where $i$ and $j$ represent $a$ and $b$. Some attention weights are omitted, such as $e_{c_2 \rightarrow c_1}$.
>
> We classify attention into two types: syntactic attention, which reflects grammatical relationships, and semantic attention, which captures meaning connections, often between paragraphs. When averaging token-level attentions, we only extract attentions between high-level and low-level nodes, omitting attentions within the same node. This process emphasizes long-distance semantic attention, which effectively reflects the semantic relationships between nodes.
>
> All calculations use efficient tensor operations, making their computational cost negligible relative to the LLM processing. Optimization includes a global variable to iteratively accumulate averaged attention weights per layer, introducing only minimal memory overhead.
>
> We have added these explanations to Appendix B in the revised version.

---

> ### Author Response · Authors · 2024-11-17
>
> ---
> > **Q2**: In Section 3.2.2, the paper discusses calculating the relevance between the query and information points using attention. How is this calculation performed, especially if it needs to be precomputed?
>
> **A2**:
> The attention values are directly extracted from the LLM as shown in Figure 3. We utilize the computed attention weights from the LLM and apply an averaging operation. This additional computation is minimal compared to the cost of running the LLM itself.
>
> For a query with tokens $[q_1, q_2]$ and a node with tokens $[t_1, t_2]$, we first average the attention weights across attention heads and layers. This results in the attention weights $e_{t_1 \rightarrow q_1}, e_{t_2 \rightarrow q_1}, e_{t_1 \rightarrow q_2}, e_{t_2 \rightarrow q_2}$, while excluding intra-node attentions like $e_{t_2 \rightarrow t_1}$.
>
> Next, we average the attention weights over the query tokens. $e_{t_1 \rightarrow q_1}$ and $e_{t_1 \rightarrow q_2}$ are averaged into $e_{t_1 \rightarrow q}$. Similarly, $e_{t_2 \rightarrow q_1}$ and $e_{t_2 \rightarrow q_2}$ are averaged into $e_{t_2 \rightarrow q}$. Finally, $e_{t_1 \rightarrow q}$ and $e_{t_2 \rightarrow q}$ are averaged to obtain $e_{t \rightarrow q}$, representing the final attention between the query and the node.
>
> From an implementation perspective, all calculations are performed using efficient tensor operations. Averaging across heads and layers introduces minimal computational overhead. Additionally, we maintain a global variable to iteratively sum attention weights for each layer when running the LLM, ensuring limited memory usage relative to the LLM processing.
>
>
> ---
> > **Q3**: This method may have limited scalability for new queries and new documents.
>
> **A3**:
>
> For **new queries**:
>
> Our method is highly adaptable to new queries. The graph construction is performed only once per document. New queries can directly leverage the constructed graph, requiring minimal computational effort.
>
> For **new documents**:
>
> We address new document issue as follows:
> 1. **Efficiency for long documents**: For lengthy texts like those in NarrativeQA, our method processes documents in manageable nodes, limiting input length to under 8K. This reduces computational demands and enables inference for long documents using a single GPU with fewer TFLOPs even for single-query documents, as shown in Table 3. In contrast, LLMs scale quadratically with sequence length, requiring significant GPU memory and compute power.
> 2. **Preprocessing vs inference**: In real-world applications, we often prioritize inference speed over the preprocessing stage. For instance, when deploying our method in a product, Graph Construction can be done in advance on a high-capacity server, allowing for efficient inference on personal local devices. Directly using an LLM for inference in such scenarios would be computationally intensive for a local device.
> 3. **Single-time preprocessing in an RAG manner**: Graph Construction only needs to be performed once per document, akin to the indexing phase in RAG. RAPTOR and MeMWalker also require substantial computation resources to build the summary trees before inference.
>
>
> ---
> > **Q4**: No code is released, making it difficult for others to use the proposed method or reproduce the result.
>
> **A4**:
> We will make the code publicly upon paper acceptance (see footnote #1 on page #1).

---

> ### Author Response · Authors · 2024-11-23
>
> Thank you again for your thoughtful and detailed feedback. We've taken your initial feedback into careful consideration and addressed each point in our responses. Could you kindly confirm whether our responses have appropriately addressed your concerns? We would greatly appreciate it if you could take our responses into account during discussions with the AC and other reviewers. Please let us know if you have further comments.
>
> Thank you once again for your time and effort in reviewing our work.

---

### Official Review · Reviewer_mYAX · 2024-10-31

**Soundness:** 3
**Presentation:** 3
**Contribution:** 3
**Rating:** 6
**Confidence:** 3

**Summary:**

This paper proposes a new retrieval method, GARLIC (LLM-Guided Dynamic Progress Control with Hierarchical Weighted Graph), which aims to improve information retrieval in long document question answering tasks. It replaces the traditional tree structure by building a hierarchical weighted directed acyclic graph and uses the attention weights of large language models (LLMs) to perform multi-path search and dynamically control the retrieval process. Experiments show that this method surpasses the existing state-of-the-art baselines, including methods that directly use Llama 3.1 to process the entire document, while maintaining the computational efficiency of the RAG method.

**Strengths:**

1. Overall, this paper is well-written and easy to read.
2. While existing RAG methods perform worse than directly feeding the entire document into Llama 3.1, the authors propose a new RAG method outperforms Llama 3.1 and retaining the computational efficiency of RAG methods. This approach can be used for reference by other researchers.

**Weaknesses:**

1. In line 292, the author mentioned that the node's attention is multiplied by the corresponding position information to track the earlier information in the sequence. I think the author should conduct experiments to explain whether other adjustment methods have been tried? Why is this adjustment method effective?
2. Lack of comparative analysis with other methods: Although the proposed model achieved excellent performance, the authors did not clearly explain why HWDAG is superior to the tree-based RAG method? Tree-based retrieval also seems to be able to summarize long documents and answer questions by selecting important nodes that are highly relevant to the query in a top-down manner through the GRAPH SEARCH method proposed by the authors. In addition, HWDAG seems to organize documents into a top-down hierarchy with different granularity, so low-level nodes seem to be a refinement and explanation of top-level nodes. As one of the main motivations of the article, I hope the author can explain why introducing refined information will hurt performance for some problems.
3. During the construction of the summary graph, the authors iteratively aggregate lower-level nodes by batching them and inputting them into the large language model (LLM) to generate higher-level nodes. However, the details of the iterative batching mechanism require further elaboration.

4. The authors argue that the information points (IPs) focus on “a single event or very few events”. It seems that the authors achieve this with just several instructions in the prompt. Is this approach sufficient to ensure the achievement of the intended objectives?

5. The authors average the token-level attentions to obtain the weights between information chunks (IPs). However, since these chunks represent higher-level, text-based content, is averaging token-level attentions an appropriate method for establishing relationships between these information chunks?

Comments, Suggestions And Typos:
1. In Section 4 Experiments, it is recommended to highlight the metrics that perform well in the table by bolding or underlining them.

**Questions:**

Please refer to weakness, thanks.

**Details Of Ethics Concerns:**

NA.

---

> ### Author Response · Authors · 2024-11-17
>
> We appreciate the reviewer's time and the thoughtful comments. The response is as follows:
>
> ---
> > **Q1**: In line 292, the author mentioned that the node's attention is multiplied by the corresponding position information to track the earlier information in the sequence. I think the author should conduct experiments to explain whether other adjustment methods have been tried? Why is this adjustment method effective?
>
> **A1**:
> We appreciate your insightful suggestion. Our chosen adjustment method is primarily heuristic. Ideally, if the input nodes are earlier in the sequence and the query appears later, there would be no need for this adjustment, as the query’s attention could naturally distribute across the nodes. However, in our setup, the query appears first and the documents follow. The adjustment allows us to cache key values for both query and documents efficiently.
>
> The intuition behind this adjustment is based on our observation that nodes later in the sequence tend to assign lower attention weights to the query. For instance, the second node splits its attention between the query, the first node, and itself, with approximately 1/3 of the attention directed towards the query. Similarly, the third node allocates roughly 1/4 of its attention to the query. To address this, we scale the attention weights of the second and third nodes by 3 and 4, respectively, to account for their position relative to the query.
>
> We acknowledge that further exploration of alternative adjustments might yield additional improvement. For this study, our goal was to focus on the overall attention-based retrieval mechanism, keeping the adjustments simple. The results show that our method achieves strong performance with this simple heuristic adjustment. We plan to explore further refinements in future work.
>
>
> ---
> > **Q2**: Lack of comparative analysis with other methods: Although the proposed model achieved excellent performance, the authors did not clearly explain why HWDAG is superior to the tree-based RAG method? Tree-based retrieval also seems to be able to summarize long documents and answer questions by selecting important nodes that are highly relevant to the query in a top-down manner through the GRAPH SEARCH method proposed by the authors.
>
> **A2**:
> Even if a tree-based RAG incorporates our Graph Search method, it would still face limitations compared to our approach. This is supported by our experimental results in the Ablation Study (Table 2 and Line 471-473), which show a performance drop when replacing the graph structure with a tree.
>
> While Dynamic Progress Control could adapt to a tree-based RAG, attention-based retrieval would not perform effectively in a tree structure. Our method relies on fine-grained events at each node. Our intuition is that if an event is relevant to the query, the model can leverage detailed information from lower-level nodes of the same event. A single low-level node may contain multiple events, and different low-level nodes may reference the same event, creating a many-to-many relationship between events and nodes, as shown in Figure 5. High-level nodes can organize the events into different nodes, with each connected to its source nodes at the lower level.
>
> In contrast, in the tree-based method, each node aggregates events from all its child nodes, with attention only indicating which child node contributes most to the parent node. This approach misses the internal semantic relationships we capture, such as how each child node contributes to different parts of the parent node.
>
> To capture these details in a tree, one would need to split parent nodes into sentence-level segments to identify how each child contributes. However, this approach actually transforms the tree into a graph. Additionally, splitting nodes directly into sentences poses challenges, as events may span multiple sentences. For example, sentences that begin with pronouns may lose referential context after split.
>
> Our method employs LLM to achieve event segmentation. Compared to tree-based search, our graph-based approach functions more as an event-tracking process, using semantic relationships through attention-based retrieval. During the search, it follows events from high-level nodes down to lower-level details, with Dynamic Progress Control halting the search once sufficient detail is reached.

---

> ### Author Response · Authors · 2024-11-17
>
> ---
> > **Q3**: In addition, HWDAG seems to organize documents into a top-down hierarchy with different granularity, so low-level nodes seem to be a refinement and explanation of top-level nodes. As one of the main motivations of the article, I hope the author can explain why introducing refined information will hurt performance for some problems.
>
> **A3**:
> Our intuition is that retrieving more refined information generally improves performance, particularly when using powerful LLMs like Llama 3.1. However, retrieving more information increases computational costs, so we aim to strike a balance between effectiveness and efficiency, depending on the specific query. The misunderstanding may arise from the role of Dynamic Progress Control in our method. Our motivation is rooted in the fact that queries vary in complexity and information needs. GARLIC adapts dynamically to query needs.
>
> ​​For example, GARLIC retrieves 2 nodes for an easy query, 5 for a medium query, and 8 for a hard query spread across different document sections. In contrast, baselines retrieve a fixed number (e.g., 5) for all queries. This adaptive retrieval enhances efficiency for simpler queries and effectiveness for harder ones, with similar average retrieval volume and computation cost.
>
> For simpler queries, refined information may be unnecessary, as the top-level nodes already provide sufficient context to answer the query. In such cases, Dynamic Progress Control stops the search early to prioritize efficiency. Conversely, for harder queries, the method extends the search to gather more refined information spread across different sections of the document.
>
> In the "Top-X" experiments (Table 1), we ensure that baselines retrieve the same or greater average volume of information as our method. Despite this, our approach achieves superior results, demonstrating its ability to balance refined retrieval and query-specific needs effectively. Additionally, the stop patience parameter can be adjusted to control the trade-off between efficiency and effectiveness.
>
>
> ---
> > **Q4**: During the construction of the summary graph, the authors iteratively aggregate lower-level nodes by batching them and inputting them into the large language model (LLM) to generate higher-level nodes. However, the details of the iterative batching mechanism require further elaboration.
>
> **A4**:
> Unlike RAPTOR, which batches lower-level nodes using clustering, we simply batch low-level nodes in sequential order. For each layer, we gather nodes into batches based on their token counts. The token count includes the nodes themselves but excludes tokens used for the prompt and the generated summary. Therefore, each batch is capped at approximately 6K tokens to ensure the total token count does not exceed the model's 8K limit.
>
> For example, first-level nodes $\{v^1_i\} _{1}^{30}$  might be divided into batches such as  $\{v^1_i\} _{1}^{10}$, $\{v^1_i\} _{11}^{20}$, and $\{v^1_i\} _{21}^{30}$, with each batch containing consecutive nodes. This approach is consistent even when processing multiple documents. The model then generates second-level nodes for each batch, e.g., $\{v^2_i\} _{1}^{15}$, $\{v^2_i\} _{16}^{20}$, and $\{v^2_i\} _{21}^{40}$. In this process, $\{v^1_i\} _{1}^{10}$ is summarized into $\{v^2_i\} _{1}^{15}$, $\{v^1_i\} _{11}^{20}$ is summarized into $\{v^2_i\} _{16}^{20}$, and so on. The number of higher-level nodes may vary depending on the semantic context of the summaries.
>
> For subsequent levels, the nodes are similarly batched, but as higher-level nodes typically contain fewer tokens, each batch can include more nodes. For instance, second-level nodes $\{v^2_i\} _{1}^{40}$ might be split into batches $\{v^2_i\} _{1}^{20}$ and $\{v^2_i\} _{21}^{40}$. This allows higher-level nodes to summarize information from a variable number of lower-level nodes, depending on the semantic context of the content.

---

> ### Author Response · Authors · 2024-11-17
>
> ---
> > **Q5**: The authors argue that the information points (IPs) focus on “a single event or very few events”. It seems that the authors achieve this with just several instructions in the prompt. Is this approach sufficient to ensure the achievement of the intended objectives?
>
> **A5**:
> Yes, we achieve this with carefully designed instructions in the prompt. During our experiments, we manually reviewed at least 30 documents across different datasets and found no exceptions: Llama 3.1 consistently followed the instructions, generating Information Points (IPs) that encapsulate a single event or very few events, as illustrated in Figure 5 in the Appendix.
>
> In the instructions, we did not explicitly use terms like “Information Point” or “event.” Instead, we instruct Llama 3.1 to summarize content in the format of “bullet points,” which we observed to work effectively. This approach likely succeeds because the concept of bullet points is widely used in human language, making it a concept that LLMs already learned robustly. Furthermore, the structure and purpose of bullet points align closely with the definition of Information Points described in our paper. Additionally, we enhance clarity by instructing the LLM to format each bullet point with a leading asterisk (“*”) and to perform summarization in the process. This ensures the output adheres to the intended format while maintaining semantic focus and brevity.
>
>
> ---
> > **Q6**: The authors average the token-level attentions to obtain the weights between information chunks (IPs). However, since these chunks represent higher-level, text-based content, is averaging token-level attentions an appropriate method for establishing relationships between these information chunks?
>
> **A6**:
> We classify attention into two types: syntactic attention, which reflects grammatical relationships (e.g., subject-verb dependencies within a sentence), and semantic attention, which captures semantic meaning connections, often between paragraphs.
>
> When averaging token-level attentions, we only extract attentions between high-level and low-level nodes, omitting attentions within the same node. This process emphasizes long-distance semantic attention, which effectively reflects the semantic relationships between nodes. For instance, if a high-level node assigns 50% of its attention to a low-level node and 50% to itself, we only extract the 50% attention to the low-level node, renormalize it, and scale it so the total sums to 1.
>
> Additionally, we acknowledge that alternative methods for handling attention could potentially yield better results. However, in this study, our primary focus was to develop an overall attention-based retrieval mechanism while keeping the details of the attention-handling process simple. Despite this simplicity, the results demonstrate that our method achieves strong performance with this simple averaging. We will leave the exploration of more advanced methods for handling attention in future work.

---

> ### Author Response · Authors · 2024-11-23
>
> Thank you again for your thoughtful and detailed feedback. We've taken your initial feedback into careful consideration and addressed each point in our responses. Could you kindly confirm whether our responses have appropriately addressed your concerns? We understand you are very busy, but we would greatly appreciate it if you could take our responses into account during discussions with the AC and other reviewers. Please let us know if you have further comments.
>
> Thank you once again for your time and effort in reviewing our work.

---

> ### Comment · Reviewer_mYAX · 2024-11-24
> **Response**
>
> Thanks for your detailed illustrations. I think the authors have addressed most of my concerns. I'll keep my scores. Thanks.

---

### Official Review · Reviewer_cbqx · 2024-11-01

**Soundness:** 3
**Presentation:** 3
**Contribution:** 2
**Rating:** 5
**Confidence:** 4

**Summary:**

The paper presents GARLIC, a retrieval method that uses Hierarchical Weighted Directed Acyclic Graphs to improve long-document question-answering (QA) tasks. Unlike traditional retrieval-augmented generation (RAG) and tree-based models, GARLIC combines a unique two-stage process - Summary Graph Construction and Dynamic Graph Search.

**Strengths:**

By using dynamic stopping and attention-weighted paths, GARLIC avoids unnecessary computation, which is common in exhaustive retrieval methods.

Outperforms benchmarks like Llama 3.1 and recent methods on NarrativeQA, Qasper, HotpotQA, and MuSiQue.

Compatible with long-context LLMs (e.g., Llama 3.1), making it feasible for single GPU execution.

**Weaknesses:**

The attention normalization approach may limit scalability to larger graphs and complex attention patterns.

While the retrieval stage is efficient, graph construction is relatively resource-intensive, especially for single-query documents.

Embedding similarity, although beneficial, introduces some redundancy when compared with pure attention-based retrieval.

**Questions:**

How does the system handle differences in event granularity? Specifically, do Information Points (IPs) for shorter events lead to inconsistencies in information distribution across the graph?

How does GARLIC scale when handling complex attention dependencies or queries that span multiple, disparate sections of a document? Is there a threshold at which the attention-guided search might encounter limitations?

Has dynamically adjusting the stop patience based on query complexity or node connectivity been considered? Could this approach optimize retrieval efficiency within larger graphs?

How does the efficiency of graph search vary in contexts with mixed single- and multi-query documents, particularly in cases with highly fragmented narratives, such as those in NarrativeQA?

GARLIC appears to extend ideas from both RAPTOR and MemWalker, borrowing sub-graph or informational-node stacking from RAPTOR and summarization from MemWalker. Could you clarify the main design distinctions and advantages of GARLIC compared to these methods (apart from using greedy search)?

Could you explain the choice to exclude LongT5 XL (Guo et al., 2022) and CoLT5 XL (Ainslie et al., 2023) as baselines (or any other baseline you feel may have given better performance similar to the base Llama3.1 on the benchmarked datasets), given their performance as open-source models? While ChatGPT’s closed-source nature makes its exclusion understandable, a rationale for omitting these alternatives would help clarify comparisons. Also, could you discuss the reported GARLIC performance alongside Table 5 in [1]?

From Table 1, it appears that a significant portion of GARLIC's performance boost is coming from the Llama-3.1 base model, as it already outperforms other baselines. How might results compare if Llama-3.1 were used with RAPTOR instead, considering RAPTOR's higher efficiency (0.5:1 on 1 datasets, and always below 1 on all remaining datasets)?

[1] Sarthi, Parth, et al. "Raptor: Recursive abstractive processing for tree-organized retrieval." arXiv preprint arXiv:2401.18059 (2024)."

---

> ### Author Response · Authors · 2024-11-17
>
> We appreciate the reviewer's time and the thoughtful comments. The response is as follows:
>
> > **Q1**: The attention normalization approach may limit scalability to larger graphs and complex attention patterns.
>
> **A1**:
> We address the scalability of attention normalization in terms of both time complexity and space complexity:
>
> **Time Complexity**:
>
> The model’s time complexity is $O(nds^2 + nsd^2)$, where $s$, $d$, and $n$ represent the sequence length, hidden size, and layer number, respectively. Attention normalization requires $O(nms^2)$, where $m$ represents the attention head number, which is negligible compared to the complexity $O(nds^2)$, even for long sequences, as $d \gg m$ in LLMs.
>
> Our method caps the sequence length $s$ to 8K, mitigating the quadratic growth of LLM computational costs with sequence length. Despite introducing an additional summarization step, the total TFLOPs used are still lower than that of Llama3.1 for long documents, such as those in the NarrativeQA dataset.
>
> **Space Complexity**:
>
> During model execution, attention is normalized layer by layer. We use a global variable, `attention_weight_layer_sum`, to accumulate normalized attention across layers, as shown in the pseudocode below:
>
> ```python
> def forward(self, input_ids, ...):
>     ...
>     attention_weight_layer_sum = 0
>     for layer in self.layers:
>         ...
>         hidden_states, attention_weight = layer(hidden_states)
>         normlized_attention_weight_along_head = normalize(attention_weight)
>         attention_weight_layer_sum += normlized_attention_weight_along_head
>         ...
>     attention_weight = normalize(attention_weight_layer_sum)
>     ...
>     return output, attention_weight, ...
> ```
>
> The global variable `attention_weight_layer_sum` accumulates normalized attention weights iteratively across layers. After each layer’s forward pass, attention is normalized along the attention head dimension to further reduce memory usage and added to `attention_weight_layer_sum`. Consequently, the model and activation space complexity remains $O(nd^2 + sd + ms^2)$, with an additional memory usage of only $O(s^2)$, which is significantly smaller than $O(ms^2)$, even for long sequences.
>
> By limiting the sequence length $s$ to 8K in our method, we further alleviate the LLM's memory growth, allowing our method to run on a single GPU for NarrativeQA, whereas Llama3.1 would require 8 GPUs.
>
> In summary, based on the time and space complexity analysis, the computation and memory demands of attention normalization are minor compared to the overall cost of the LLM, even for long sequences. The primary scalability bottleneck remains within the LLM itself.
>
>
> ---
> > **Q2**: While the retrieval stage is efficient, graph construction is relatively resource-intensive, especially for single-query documents.
>
> **A2**:
> We address the resource-intensiveness of Graph Construction as follows:
> 1. **Efficiency for long documents:** For lengthy texts like those in NarrativeQA, our method processes documents in manageable nodes, limiting input length to under 8K. This reduces computational demands and enables inference for long documents using a single GPU with fewer TFLOPs even for single-query documents, as shown in Table 3. In contrast, LLMs scale quadratically with sequence length, requiring significant GPU memory and compute power.
> 2. **Preprocessing vs inference:** In real-world applications, we often prioritize inference speed over the preprocessing stage. For instance, when deploying our method in a product, Graph Construction can be done in advance on a high-capacity server, allowing for efficient inference on personal local devices. Directly using an LLM for inference in such scenarios would be computationally intensive for a local device.
> 3. **Single-time preprocessing in an RAG manner:** Graph Construction only needs to be performed once per document, akin to the indexing phase in RAG. RAPTOR and MeMWalker also require substantial computation resources to build the summary trees before inference.

---

> ### Author Response · Authors · 2024-11-17
>
> ---
> > **Q3**: Embedding similarity, although beneficial, introduces some redundancy when compared with pure attention-based retrieval.
>
> **A3**:
> We consider this a strength of our work. In this paper, we propose a new attention-based retrieval paradigm that goes beyond traditional embedding similarity. While incorporating embedding similarity add some redundancy, as shown in Table 2, removing it results in only a minor performance drop. This demonstrates the robustness of our proposed attention-based retrieval: even without relying on embedding similarity used in previous baselines, our method maintains strong performance.
>
>
> ---
> > **Q4**: How does the system handle differences in event granularity? Specifically, do Information Points (IPs) for shorter events lead to inconsistencies in information distribution across the graph?
>
> **A4**:
> Information Points (IPs) create intentional differences in information distribution between high-level and low-level nodes, which is a key feature of our method.
>
> In our graph structure, high-level nodes represent events in a more condensed form, capturing essential details, while low-level nodes provide a more detailed account of the same events.
>
> For example, let's say we have two events, Event #1 and Event #2: In a lower-level (more detailed) node, Event #1 might be described with 100 words, and Event #2 with 50 words. In a higher-level (less detailed) node, both Event #1 and Event #2 may be condensed to around 10 words each, focusing on key information, such as "who did what," while omitting finer details.
>
> This intentional design ensures that high-level nodes prioritize uniform representation across events, avoiding situations where events with fewer details are disproportionately underrepresented. If high-level nodes were to maintain the same proportional information distribution as low-level nodes, events with fewer tokens might disappear at higher levels, while information-rich events would retain excessive detail. This imbalance could hinder retrieval, especially for less-detailed events.
>
> Our approach ensures that even if details are omitted for information-rich events at higher levels, the model can retrieve more detailed information by following paths to corresponding low-level nodes. This hierarchical representation enhances the retrieval process by maintaining a balance between efficiency and granularity.
>
>
> ---
> > **Q5**: How does GARLIC scale when handling complex attention dependencies or queries that span multiple, disparate sections of a document? Is there a threshold at which the attention-guided search might encounter limitations?
>
> **A5**:
> Our method is designed specifically to handle complex dependencies that span multiple, disparate sections of a document.
>
> As illustrated in Figure 2c, our graph-based method, supported by Dynamic Progress Control, enables searches along various paths, with each path potentially ending in different sections of the document. In contrast, prior tree-based approaches, as shown in Figure 2b, follow a single path from the root to one leaf, retrieving information from only one section.
>
> The number of graph search paths can also be dynamically adjusted based on the specific query and document. For queries that span multiple sections, the graph is capable of exploring more paths, and vice versa. Once sufficient information has been gathered, Dynamic Progress Control terminates the search process.

---

> ### Author Response · Authors · 2024-11-17
>
> ---
> > **Q6**: Has dynamically adjusting the stop patience based on query complexity or node connectivity been considered? Could this approach optimize retrieval efficiency within larger graphs?
>
> **A6**:
> This could be an interesting idea for future exploration. In our method, we rely on the LLM to make the decision on when to stop, so it’s possible that the LLM already considered the complexity of the query or node when deciding to terminate the search.
>
> In our experiments, we observed variability in the number of retrieved nodes across datasets, but the optimal stop patience consistently hovered around 5, as shown in Figure 4. This suggests that, while retrieval depth may vary by dataset, stop patience appears to correlate more closely with the LLM, with the LLM terminating only when sufficiently confident. This would make stop patience a stable hyperparameter across datasets.
>
> Nonetheless, dynamically adjusting stop patience based on query complexity could be a valuable direction for future research to further explore.
>
>
> ---
> > **Q7**: How does the efficiency of graph search vary in contexts with mixed single- and multi-query documents, particularly in cases with highly fragmented narratives, such as those in NarrativeQA?
>
> **A7**:
> As shown in the experiments in Section 4.2, our method achieves the highest improvement on NarrativeQA among the four datasets compared to other baselines, demonstrating its efficiency in handling documents with fragmented narratives like those in NarrativeQA.
>
> Our approach constructs the graph for a document one time, allowing all queries to reuse this graph. For each query, as illustrated in Figure 2c, graph search occurs along multiple paths, with the paths and depths dynamically adjusted according to the specific query. This flexibility enables our method to efficiently manage mixed single- and multi-query contexts, even within highly fragmented narratives.

---

> ### Author Response · Authors · 2024-11-17
>
> ---
> > **Q8**: GARLIC appears to extend ideas from both RAPTOR and MemWalker, borrowing sub-graph or informational-node stacking from RAPTOR and summarization from MemWalker. Could you clarify the main design distinctions and advantages of GARLIC compared to these methods (apart from using greedy search)?
>
> **A8**:
> We outline the main distinctions of GARLIC as follows:
> 1. **Dynamic Progress Control**: Traditional RAG approaches set a fixed retrieval volume, retrieving a predetermined number of nodes. Tree-based methods like RAPTOR and MemWalker follow a single path from root to leaf. GARLIC adapts dynamically to query needs. For example, it retrieves 2 nodes for an easy query, 5 for a medium query, and 8 for a hard query spread across different document sections. In contrast, baselines retrieve a fixed number (e.g., 5) for all queries. This adaptive retrieval enhances efficiency for simpler queries and effectiveness for harder ones, with similar average retrieval volume and computation cost. Dynamic Progress Control, combined with our graph-based search mechanism, also allows the model to adjust the number of paths explored based on whether a query’s information is concentrated in one chunk or dispersed across the document. In the "Top-X" experiments (Table 1), we ensure that baselines retrieve the same or greater average volume of information as our method. Despite this, our approach achieves superior results, demonstrating its ability to balance refined retrieval and query-specific needs effectively.
> 2. **New Attention-Based Retrieval Paradigm**: Unlike previous RAG methods, including RAPTOR, which rely on embedding similarity, our approach introduces a retrieval mechanism based solely on attention, without using sentence embeddings. Attention from the LLM captures node-to-node and query-to-node semantic relationships. Experiments in Table 1 and the ablation study in Table 2 demonstrate the effectiveness of this attention-based retrieval. The combination use of attention and embedding similarity further enhances performance.
> 3. **Many-to-Many Summarization for Graph Construction**: Our method constructs a graph based on many-to-many summarization, as illustrated in Figure 1, prompting the LLM to focus on extracting and connecting events across levels. An example is shown in Figure 5 in the Appendix. Lower-level nodes capture finer details, while high-level nodes provide concise summaries, allowing efficient tracking of events across document sections. For queries that involve multiple events across sections, our method efficiently tracks events through different paths to different leaves, retrieving details dispersed throughout the document. When multiple events co-occur in a single chunk, our approach captures the inter-event relations with higher retrieval scores for the relevant chunk. In the ablation study (Table 2), we observe a performance drop when replacing the graph structure with a tree, highlighting the importance of this design.

---

> ### Author Response · Authors · 2024-11-17
>
> ---
> > **Q9**: Could you explain the choice to exclude LongT5 XL (Guo et al., 2022) and CoLT5 XL (Ainslie et al., 2023) as baselines (or any other baseline you feel may have given better performance similar to the base Llama3.1 on the benchmarked datasets), given their performance as open-source models? While ChatGPT’s closed-source nature makes its exclusion understandable, a rationale for omitting these alternatives would help clarify comparisons.
>
> **A9**:
> We consider Llama3.1 for the following reasons:
> 1. In our experiments, all baselines and GARLIC use the same base model, Llama3.1. Keeping the same LLM across all methods ensures a fair comparison, focusing on the retrieval mechanism rather than variations in LLM architecture.
> 2. LongT5 XL and CoLT5 XL support input lengths of 16K and 64K tokens, respectively, while Llama3.1 supports an input length of up to 128K tokens. This extended capacity is particularly beneficial for handling long documents, such as those in NarrativeQA. By using Llama3.1 across methods, we aim to reduce the impact of input length limitations and provide a fairer comparison. While we limited our method to an 8K input length, which ensures compatibility with various LLMs, we did not impose this restriction on direct queries to the LLM (Llama3.1).
> 3. The Llama series is among the most widely used open-source models in both academia and industry. LongT5 XL and CoLT5 XL require fine-tuning on domain-specific data as they are not aligned to human preferences via RLHF or DPO, while Llama is capable of zero-shot responses with instruction-following capabilities. Using the latest version of the Llama series, Llama3.1 (as of our submission date), aligns with common practices and provides more relevant insights for a broader audience, particularly those using or considering Llama.
>
>
> ---
> > **Q10**: Also, could you discuss the reported GARLIC performance alongside Table 5 in [1]?
> [1] Sarthi, Parth, et al. "Raptor: Recursive abstractive processing for tree-organized retrieval." arXiv preprint arXiv:2401.18059 (2024).
>
> **A10**:
>
> * **Original RAPTOR setup**: RAPTOR originally uses ChatGPT-3.5 for summarization to construct the summary tree and employs LongT5 XL, CoLT5 XL, and GPT-4 for answering queries based on the retrieved nodes. LongT5 XL and CoLT5 XL are not aligned with human preferences through RLHF or DPO and thus require fine-tuning on domain-specific data. GPT-4, which is notably stronger, is used in zero-shot mode for query responses.
>
> * **Our setup for RAPTOR**: In our experiments with RAPTOR, we use Llama3.1-8B for both summarization and query answering based on retrieved nodes. To answer queries, we prompt Llama3.1 in zero-shot mode, following a similar approach to how RAPTOR originally prompted GPT-4.
>
> In conclusion, the original RAPTOR uses ChatGPT-3.5 for summarization, while we rerun RAPTOR using Llama3.1-8B. RAPTOR originally uses LongT5 XL and CoLT5 XL that are fine-tuned on domain-specific data, whereas we use Llama3.1 in a zero-shot manner for query responses in RAPTOR experiments. Although RAPTOR also uses GPT-4 in the zero-shot setting, GPT-4 remains a stronger model.
>
> Considering these adjustments, the results would be more interpretable. We ensured that all baselines and GARLIC use the same model, Llama3.1, for both summarization and answering, to provide a fairer comparison (see Line 344).
>
>
> ---
> > **Q11**: From Table 1, it appears that a significant portion of GARLIC's performance boost is coming from the Llama-3.1 base model, as it already outperforms other baselines. How might results compare if Llama-3.1 were used with RAPTOR instead, considering RAPTOR's higher efficiency (0.5:1 on 1 datasets, and always below 1 on all remaining datasets)?
>
> **A11**:
> We used the same model, Llama-3.1, across all baselines, including RAPTOR, for both summarization and query answering (see Line 344). The results are available in Table 1. This was implemented by running the RAPTOR source code with Llama-3.1. This experimental setup ensures a fair comparison by minimizing the influence of different LLM choices.

---

> ### Author Response · Authors · 2024-11-23
>
> Thank you again for your thoughtful and detailed feedback. We've taken your initial feedback into careful consideration and addressed each point in our responses. Could you kindly confirm whether our responses have appropriately addressed your concerns? We would greatly appreciate it if you could take our responses into account during discussions with the AC and other reviewers. Please let us know if you have further comments.
>
> Thank you once again for your time and effort in reviewing our work.

---

> ### Comment · Reviewer_cbqx · 2024-11-25
>
> I appreciate your thorough feedback, however, I've decided to maintain my current scoring for this evaluation.

---

> > ### Author Response · Authors · 2024-11-25
> >
> > Thank you for taking the time to read through our response.
> >
> > We added experiments of “Top-X” setting for RAPTOR. We applied the Top-X setting to the Collapsed Tree variant of RAPTOR, extracting Top-20, Top-42, Top-12, and Top-22 nodes for NarrativeQA, Qasper, HotpotQA, and MuSiQue, respectively, to match the TFLOPs of our method, consistent with other Top-X settings. RAPTOR has two variants: Tree Traversal (TT) and Collapsed Tree (CT). Collapsed Tree collapses the tree into a single layer and retrieves nodes until a threshold is reached, based on embedding similarity to the query vector.
> >
> > Detailed RAPTOR results are provided below:
> >
> > | NarrativeQA | F1 | ROUGE-L | BLEU-4 | TFLOPs | Ratio |
> > |---|---|---|---|---|---|
> > |Llama3.1 | 53.7 | 52.6 | 10.4 | 3361.9 | 108.45x |
> > |RAPTOR-TT | 40.6 | 39.8 | 7.8 | 20.3 | 0.65x |
> > |RAPTOR-CT | 48.6 | 47.8 | 11.8 | 17.9 | 0.58x |
> > |RAPTOR-CT Top-X | 52.0 | 51.2 | 11.8 | 35.1 | 1.13x |
> > |GARLIC | 61.1 | 60.2 | 18.6 | 31.0 | 1.00x |
> >
> > | Qasper | F1 | ROUGE-L | BLEU-4 | TFLOPs | Ratio |
> > |---|---|---|---|---|---|
> > |Llama3.1 | 49.4 | 47.6 | 26.9 | 92.5 | 1.38x |
> > |RAPTOR-TT | 42.1 | 40.1 | 17.2 | 17.7 | 0.26x |
> > |RAPTOR-CT | 44.6 | 42.7 | 19.5 | 16.6 | 0.25x |
> > |RAPTOR-CT Top-X | 46.9 | 44.7 | 20.8 | 67.3 | 1.01x |
> > |GARLIC | 49.7 | 47.9 | 27.0 | 66.9 | 1.00x |
> >
> > | HotpotQA | F1 | ROUGE-L | BLEU-4 | TFLOPs | Ratio |
> > |---|---|---|---|---|---|
> > |Llama3.1 | 41.3 | 41.2 | 6.3 | 23.7 | 1.48x |
> > |RAPTOR-TT | 38.6 | 38.5 | 6.7 | 8.4 | 0.53x |
> > |RAPTOR-CT | 40.9 | 40.4 | 7.2 | 15.3 | 0.96x |
> > |RAPTOR-CT Top-X | 40.7 | 40.7 | 7.2 | 17.9 | 1.12x |
> > |GARLIC | 43.5 | 43.5 | 7.2 | 16.0 | 1.00x |
> >
> > | MuSiQue | F1 | ROUGE-L | BLEU-4 | TFLOPs | Ratio |
> > |---|---|---|---|---|---|
> > |Llama3.1 | 35.8 | 35.7 | 5.6 | 40.6 | 1.31x |
> > |RAPTOR-TT | 29.3 | 29.3 | 4.7 | 12.6 | 0.41x |
> > |RAPTOR-CT | 31.5 | 31.5 | 5.5 | 16.1 | 0.52x |
> > |RAPTOR-CT Top-X | 35.4 | 35.2 | 7.2 | 32.2 | 1.04x |
> > |GARLIC | 36.9 | 36.8 | 5.7 | 30.9 | 1.00x |
> >
> > RAPTOR-CT outperforms RAPTOR-TT, consistent with the conclusions in the RAPTOR paper. The top-to-bottom search strategy of RAPTOR-TT does not capture sufficient information to effectively answer queries. RAPTOR-CT Top-X demonstrates better performance than the original RAPTOR when provided with more nodes and TFLOPs.
> >
> > Our method still consistently outperforms RAPTOR-CT Top-X across all four datasets, even under similar TFLOPs, with particularly strong results on the long documents from NarrativeQA. Under this setting, RAPTOR-CT Top-X and our method exhibit similar computational costs in both the summary and inference stages, as both rely on a hierarchical summary structure. Thus, RAPTOR is also “relatively resource-intensive, especially for single-query documents,” as pointed out in the second weakness. This also shows that our method has superior efficiency and resource utilization under the similar computational rescources. For example, as noted in Q8/A8.1, our Dynamic Progress Control mechanism dynamically allocates fewer resources to simpler queries and more to complex ones, enhancing overall performance.
> >
> > Our method consistently outperforms RAPTOR-CT Top-X across all four datasets, even under similar TFLOPs. In this setting, RAPTOR-CT Top-X and our method have comparable computational costs in both the summary and inference stages, as both utilize a hierarchical summary structure. Consequently, as noted in the second weakness, RAPTOR is also “relatively resource-intensive, especially for single-query documents.” These results further demonstrate our method's superior efficiency and resource utilization under similar computational resources. Our Dynamic Progress Control, as highlighted in Q8/A8.1 effectively allocates fewer resources to simpler queries and more to complex ones, leading to enhanced overall performance.
> >
> > Importantly, with the same TFLOPs, RAPTOR-CT Top-X still underperforms Llama 3.1. Among all baselines, our method is the only one that outperforms Llama 3.1 across all datasets, underscoring the unique effectiveness of our approach.
> >
> > When comparing with Llama 3.1, the computational cost of attention normalization is negligible as noted in Q1/A1. Furthermore, the resource-intensive nature for single-query documents, highlighted in the second weakness, is also mitigated when comparing with Llama 3.1. For instance, for a single-query document from NarrativeQA, Llama 3.1 requires 3361.9 TFLOPs, whereas our method only requires 2073.8 TFLOPs, encompassing both the summary and inference stages. Moreover, for multi-query documents, our method is significantly more efficient than Llama 3.1.
> >
> > If we have adequately addressed all your concerns, we kindly ask if you would consider updating your score. If there are any remaining concerns or reasons for you keeping your score, please let us know so we can make further changes. Thank you for your time and consideration.

---

### Official Review · Reviewer_kpAv · 2024-11-02

**Soundness:** 2
**Presentation:** 2
**Contribution:** 2
**Rating:** 3
**Confidence:** 3

**Summary:**

This work introduces a retrieval method, "GARLIC" (LLM-Guided Dynamic Progress Control with Hierarchical Weighted Graph), aimed at improving long document question answering. GARLIC operates through a multi-step process. First, it constructs a Hierarchical Weighted Directed Acyclic Graph (HWDAG) where each node represents a focused "Information Point" (IP) generated through LLM-based summarization. Second, GARLIC dynamically retrieves nodes from the graph, leveraging attention weights to assess relevance and terminate the search once sufficient information is collected. Third, it employs a Greedy Best-First Search to explore multiple graph paths, enhancing flexibility by allowing the search to stop at various levels based on the query's needs. This approach effectively balances retrieval precision and computational efficiency, outperforming existing methods in single and multi-document QA tasks.

**Strengths:**

S1. The proposed method employs a hierarchical weighted directed acyclic graph instead of a tree structure, utilizing multi-path dynamic retrieval and hierarchical summary nodes.
S2. It is an interesting idea to assign weights to the edges based on attention, which allows GARLIC to adjust the retrieval depth and information volume flexibly.
S3. Decent performance compared to other existing approaches for retrieval.

**Weaknesses:**

W1. An explanation of the edge weights in the directed acyclic graph needs further clarification: specifically, how the weights are calculated and how they function during the search process. The current presentation lacks intuitiveness.
W2. Detailed explanation on how KV caching is applied in the dynamic progress control is needed.
W3. The code should be made publicly available.
W4. Some baseline methods mentions in the article search down to the leaf nodes during retrieval, in other words, gather more information. However, why do they perform worse than the method presented in this paper, even though the proposed method may terminate the search early? Further analysis would be beneficial.

**Questions:**

See weakness

---

> ### Author Response · Authors · 2024-11-17
>
> We appreciate the reviewer's time and the thoughtful comments. The response is as follows:
>
> ---
> > **Q1**: W1. An explanation of the edge weights in the directed acyclic graph needs further clarification: specifically, how the weights are calculated.
>
> **A1**:
> Given input tokens such as $[a_1, a_2], [b_1, b_2]$ for nodes $a$ and $b$, and output tokens $[c_1, c_2], [d_1, d_2]$ for nodes $c$ and $d$, edge weights are calculated as follows:
>
> 1. The LLM generates many-to-many summaries from $[a_1, a_2], [b_1, b_2]$ to $[c_1, c_2], [d_1, d_2]$.
> 2. During processing, attention scores are recorded and averaged across all attention heads and layers. For example, attention weights $e_{c_1 \rightarrow b_1}$ across all heads and layers are averaged into a single attention weight $e_{c_1 \rightarrow b_1}$ for the summary.
> 3. Token-level attention weights are grouped and averaged by node. For example, attention weights $e_{c_1 \rightarrow b_1}$ and $e_{c_1 \rightarrow b_2}$ are averaged to yield a single weight $e_{c_1 \rightarrow b}$. This is repeated for all input nodes:  $e_{c_1 \rightarrow a}$, $e_{c_2 \rightarrow a}$, $e_{d_1 \rightarrow a}$, $e_{d_2 \rightarrow a}$, $e_{c_1 \rightarrow b}$, $e_{c_2 \rightarrow b}$, $e_{d_1 \rightarrow b}$, $e_{d_2 \rightarrow b}$.
> 4. Similarly, for output $d$, we average $e_{c_1 \rightarrow b}$ and $e_{c_2 \rightarrow b}$ into $e_{c \rightarrow b}$. This step is repeated for all output nodes: $e_{c \rightarrow a}$, $e_{d \rightarrow a}$, $e_{c \rightarrow b}$,  $e_{d \rightarrow b}$.
> 5. Finally,  output-level weights are normalized to sum to 1. For example, for output $d$, $e_{c \rightarrow a}$, and $e_{c \rightarrow b}$, are normalized to sum to 1, using $e_{c \rightarrow i} = \frac{e_{c \rightarrow i}}{\sum e_{c \rightarrow j}}$, where $i$ and $j$ represent $a$ and $b$. Some attention weights are omitted, such as $e_{c_2 \rightarrow c_1}$.
>
> We classify attention into two types: syntactic attention, which reflects grammatical relationships, and semantic attention, which captures meaning connections, often between paragraphs. When averaging token-level attentions, we only extract attentions between high-level and low-level nodes, omitting attentions within the same node. This process emphasizes long-distance semantic attention, which effectively reflects the semantic relationships between nodes.
>
> All calculations use efficient tensor operations, making their computational cost negligible relative to the LLM processing. Optimization includes a global variable to iteratively accumulate averaged attention weights per layer, introducing only minimal memory overhead.
>
> We have added these explanations to Appendix B in the revised version.

---

> ### Author Response · Authors · 2024-11-17
>
> ---
> > **Q2**: W1. how they (attention weights) function during the search process. The current presentation lacks intuitiveness.
>
> **A2**:
> The intuition is that a high-level node closely related to the query will likely retrieve its detailed successors.  Once sufficient related details are retrieved, the Dynamic Progress Control terminates the search.
>
> This involves two measures:
> 1. **Query-node relatedness** (attention weights during the search, Lines 287–289).
> 2. **Node-successor relatedness** (extracted during Graph Construction via summarization, as described in Lines 241–245; further elaboration has been added to Appendix B, as mentioned in the A1 response).
>
> The final retrieval score for a node is the product of these two relatedness values. Thus, a node will be retrieved only if it contains details linked to a visited node that is highly relevant to the query.
>
> For example, consider a many-to-many summary $[a, b] \rightarrow [c, d]$:
> * If the high-level node $c$ is visited and highly related to the query (relatedness score $q_c$), we aim to retrieve details related to $c$. Attention weights $e_{c \rightarrow a}$ and $e_{c \rightarrow b}$ determine  $c$’s reliance on $a$ or $b$. If $c$ attends more to $a$, $a$ is more likely to be retrieved, with a retrieval score of $q_c e_{c \rightarrow a}$ for $a$ and $q_c e_{c \rightarrow b}$ for $b$.
> * If both $c$ and $d$ are visited, we consider all relevant weights: $e_{c \rightarrow a}$, $e_{c \rightarrow b}$, $e_{d \rightarrow a}$, and $e_{d \rightarrow b}$. The retrieval scores for $a$ and $b$ then become $q_c e_{c \rightarrow a} + q_d e_{d \rightarrow a}$ and $q_c e_{c \rightarrow b} + q_d e_{d \rightarrow b}$, respectively.
>
> This process adapts dynamically. If $c$ is highly related to the query (high $q_c$) and $d$ is not, the method focuses on nodes related to $c$. If both $c$ and $d$ are relevant to the query, the method considers all corresponding weights, retrieving nodes connected to both.
>
> The example provided is a segment of the graph, including only one set of many-to-many summarization within the entire graph structure. In practice, all many-to-many summarizations are considered, and nodes with the highest retrieval scores across all subgraphs are retrieved.
>
> All calculations use the graph’s adjacency matrix, as shown in Section 3.2.2 Line 297, which allows for efficient matrix operations, keeping computational overhead minimal compared to the LLM’s processing.

---

> ### Author Response · Authors · 2024-11-17
>
> ---
> > **Q3**: W2. Detailed explanation on how KV caching is applied in the dynamic progress control is needed.
>
> **A3**:
> KV caching refers to storing the attention key and value tensors for each layer during decoding, which is termed as "KV cache" in [1] and inspired many following works in different scenarios, such as memory management [2,3], cache merging [4,5], and compression [6,7]. The main idea of  KV caching is a mechanism used in transformer architectures to store and reuse the computed key ($K$) and value ($V$) states of previously processed nodes. This caching avoids redundant computations, particularly in iterative processes. In the context of KV caching, $K$ and $V$ are stored in the cache for previously retrieved nodes. This means that for a new input node, only its states need to be computed, while $K$ and $V$ from earlier nodes are reused from the cache.
>
> As shown in Figure 3, all key-value (KV) pairs from previously retrieved nodes are cached. For instance, consider an initial KV cache of $[q, v_1, v_2]$, where $q$ denotes the query and $v$ represents nodes. If the method retrieves a new node $v_3$, the current state consists of the KV cache $[q, v_1, v_2]$ and input $[v_3]$. During attention computation $softmax(QK^T)V$, the query states in $Q$ correspond only to the input $v_3$, while the keys ($K$) and values ($V$) include the cached states from $q$, $v_1$, and $v_2$, which were computed earlier and stored in the KV cache. This ensures that the attention mechanism can integrate information from both the newly retrieved node and previously cached nodes efficiently. We pass the KV cache and input to the model, which then outputs a response (Yes or No) along with the updated KV cache $[q, v_1, v_2, v_3]$.
>
> Suppose the next retrieved node is $v_4$. The KV cache is now $[q, v_1, v_2, v_3]$ with input $[v_4]$. Again, the query states in $Q$ pertain only to $v_4$, while the keys and values incorporate the cached states from all prior nodes $[q, v_1, v_2, v_3]$. Passing these through the model results in a response and an updated cache $[q, v_1, v_2, v_3, v_4]$. This iterative process ensures that the attention mechanism dynamically integrates information from previously retrieved nodes while maintaining computational efficiency.
>
> This approach is easily implemented with HuggingFace, as illustrated in the pseudocode below:
>
> ```python
> while search_not_end:
>     ...
>     retrieved_node = graph_search(...)
>     intput_ids, ... = process(retrieved_node, ...)
>     response, current_key_values, ... = model(input_ids=intput_ids, past_key_values=past_key_values, ...)
>     ...
>     past_key_values = current_key_values
>     ...
> ```
>
> We have added these explanations to Appendix C in the revised version.
>
>
> [1] Reiner Pope, Sholto Douglas, Aakanksha Chowdhery, Jacob Devlin, James Bradbury, Anselm Levskaya, Jonathan Heek, Kefan Xiao, Shivani Agrawal, Jeff Dean. Efficiently Scaling Transformer Inference. arXiv preprint arXiv:2211.05102 (2022).
>
> [2] Woosuk Kwon, Zhuohan Li, Siyuan Zhuang, Ying Sheng, Lianmin Zheng, Cody Hao Yu, Joseph Gonzalez, Hao Zhang, Ion Stoica. Efficient Memory Management for Large Language Model Serving with PagedAttention. SOSP 2023.
>
> [3] Wonbeom Lee, Jungi Lee, Junghwan Seo, Jaewoong Sim.InfiniGen: Efficient Generative Inference of Large Language Models with Dynamic KV Cache Management. OSDI 2024.
>
> [4] Jiayi Yao, Hanchen Li, Yuhan Liu, Siddhant Ray, Yihua Cheng, Qizheng Zhang, Kuntai Du, Shan Lu, Junchen Jiang. CacheBlend: Fast Large Language Model Serving for RAG with Cached Knowledge Fusion. SIGCOMM 2024.
>
> [5] Jang-Hyun Kim, Junyoung Yeom, Sangdoo Yun, Hyun Oh Song. Compressed Context Memory For Online Language Model Interaction. ICLR 2024.
>
> [6] Yuhan Liu, Hanchen Li, Yihua Cheng, Siddhant Ray, Yuyang Huang, Qizheng Zhang, Kuntai Du, Jiayi Yao, Shan Lu, Ganesh Ananthanarayanan, Michael Maire, Henry Hoffmann, Ari Holtzman, Junchen Jiang. CacheGen: KV Cache Compression and Streaming for Fast Large Language Model Serving. SIGCOMM 2024.
>
> [7] Suyu Ge, Yunan Zhang, Liyuan Liu, Minjia Zhang, Jiawei Han, Jianfeng Gao. Model Tells You What to Discard: Adaptive KV Cache Compression for LLMs. ICLR 2024.

---

> ### Author Response · Authors · 2024-11-17
>
> ---
> > **Q4**: W3. The code should be made publicly available.
>
> **A4**:
> We will make the code publicly upon paper acceptance (see footnote #1 on page #1).
>
>
> ---
> > **Q5**: W4. Some baseline methods mentions in the article search down to the leaf nodes during retrieval, in other words, gather more information. However, why do they perform worse than the method presented in this paper, even though the proposed method may terminate the search early? Further analysis would be beneficial.
>
> **A5**:
> In summary, our method adjusts dynamically, terminating earlier for simpler queries and later for more complex ones compared to baselines.
>
> The advantages of our approach regarding the search termination can be summarized below:
> 1. **Flexible graph exploration:** As illustrated in Figure 2c, our approach explores multiple paths in the graph, adapting the number of paths and nodes retrieved according to the query and document. In contrast, baselines search along only a single path, as shown in Figure 2b.
> 2. **Adaptive retrieval volume:** Our method retrieves information dynamically, tailored to query complexity. In contrast, traditional methods require a pre-set retrieval volume, often using a fixed number of chunks or traversing one path from the root to the leaf nodes, regardless of query complexity. For instance, given three queries:
>     * Easy queries $q_{easy}$ may need only 2 nodes;
>     * Medium queries $q_{medium}$ require 5 nodes; and
>     * Hard queries $q_{hard}$, might need 8 nodes potentially spread across different document parts.
>
>     While baselines retrieve the same fixed number of nodes (e.g., 5) for all queries, our approach retrieves 2 for $q_{easy}$, 5 for $q_{medium}$, and 8 for $q_{hard}$. This adaptive retrieval enhances efficiency for simple queries and improves effectiveness for complex ones such as $q_{hard}$, without increasing average retrieval volume or computation cost.
>
> In the "Top-X" experiments (Table 1), we ensure that baselines retrieve the same or greater average volume of information than our method. Despite this, our approach achieves superior results, demonstrating its ability to balance efficiency and query-specific retrieval effectively through Dynamic Progress Control.

---

> ### Author Response · Authors · 2024-11-23
>
> Thank you again for your thoughtful and detailed feedback. We've taken your initial feedback into careful consideration and addressed each point in our responses. Could you kindly confirm whether our responses have appropriately addressed your concerns? We understand you are very busy, but we would greatly appreciate it if you could take our responses into account during discussions with the AC and other reviewers. Please let us know if you have further comments.
>
> Thank you once again for your time and effort in reviewing our work.

---

> > ### Comment · Reviewer_kpAv · 2024-11-24
> >
> > Thanks for your response. I will keep the score.

---

> > > ### Author Response · Authors · 2024-11-24
> > >
> > > Thank you for taking the time to read through our response. May we kindly ask if our responses have fully addressed your concerns? If there is anything unclear or confusing in our responses or the updated manuscript, we would greatly appreciate the opportunity to engage in a deeper discussion. Thank you for your time.

---

> > > > ### Comment · Reviewer_kpAv · 2024-11-25
> > > >
> > > > Thanks for your response. Upon review, I believe that the contributions made by this paper are limited and cannot meet the standard of ICLR. Furthermore, I have not encountered any novel or compelling solutions to the RAG problem within the paper. So, I keep the score.

---

> ### Author Response · Authors · 2024-11-25
>
> Thank you for your valuable feedback on our work.
>
> To the best of our knowledge, we are the first to propose a **Dynamic Progress Control Mechanism** and an **Attention-Based Retrieval Paradigm** in the field of RAG. These contributions are as follows:
> * **Dynamic Progress Control**:  In existing RAG methods, the volume of information to retrieve is typically pre-defined by a hyperparameter, such as the number of chunks to retrieve in traditional methods or the number of paths from top to bottom in a summary tree for recent approaches. However, determining the optimal volume of information to retrieve is challenging due to several factors:
>     * The required information volume varies across datasets, documents, and even individual queries. Even queries within the same document may demand different levels of information. Existing methods lack the capability to determine an optimal retrieval volume for each individual query. In contrast, **Dynamic Progress Control** adaptively and dynamically determines the retrieval volume for each query.
>     * Hyperparameter tuning for retrieval volume (e.g., top-5, top-10, top-15) is computationally expensive. Our approach eliminates this overhead by adaptively setting this value (e.g., top-12, top-13, top-14) during retrieval without additional computation or hyperparameter searches.
>
>     Dynamic Progress Control is a novel mechanism that provides precise control over retrieval volume, achieving a superior balance between effectiveness and efficiency, which previous methods could not accomplish.
>
> * **Attention-based Retrieval**: Existing RAG methods predominantly rely on embedding similarity between sentences and queries for retrieval. We introduce a fundamentally different paradigm: Attention-Based Retrieval, using a many-to-many summarization graph. Our approach leverages attention mechanisms instead of embedding similarity. As demonstrated in Section 4.4, even with straightforward attention collection and normalization, our approach achieves superior results compared to embedding similarity. Attention-based retrieval captures both document-specific and query-specific latent semantic relationships, making it a promising direction for future research. This paradigm represents a novel retrieval framework with significant potential for further exploration.
>
> * **Experimental Results**: Among all baselines, our method is the only RAG approach that outperforms the LLM itself. As LLMs grow stronger and support longer input lengths, many RAG methods struggle to match the performance of LLMs, resulting in a widening performance gap. By deeply integrating the LLM into the retrieval process, our method achieves superior results, outperforming all baselines, including the LLM.
>
> We believe that **Dynamic Progress Control** and **Attention-Based Retrieval** represent valuable contributions to the community. Both are novel, previously unexplored approaches, and their effectiveness is demonstrated through our experimental results.
>
> Once again, thank you for your time and effort in reviewing our work. If possible, we would greatly appreciate it if you could share any references to existing methods that you feel closely align with the contributions we present. This would help us better understand how our work compares to prior research and identify any potential areas for improvement.

---

### Author Response · Authors · 2024-11-25
**General Response**

We sincerely thank the reviewers for their thoughtful feedback and the time they dedicated to evaluating our work.

We added the Top-X results for RAPTOR. We applied the Top-X setting to the RAPTOR Collapsed Tree variant, extracting Top-20, Top-42, Top-12, and Top-22 nodes for NarrativeQA, Qasper, HotpotQA, and MuSiQue, respectively, to match the TFLOPs of our method, consistent with other Top-X settings. The Top-X setting is to adjust the retrieved nodes to match the TFLOPs of baselines and our methods. All changes have been incorporated into the revised version and are highlighted in blue.

In the Top-X setting, we use our method to determine the nodes required for baselines, while in practice, baselines have to treat the number of extracted nodes as a hyperparameter. In contrast, our method automatically determines the number of nodes in each run using Dynamic Progress Control.

To emphasize the Top-X results and better illustrate the findings, we reordered Table 1 to list all Top-X results together.

| NarrativeQA | F1 | ROUGE-L | BLEU-4 | TFLOPs | Ratio |
|---|---|---|---|---|---|
|Llama3.1 | 53.7 | 52.6 | 10.4 | 3361.9 | 108.45x |
|MeMWalker | 11.2 | 9.8 | 2.6 | 353.8 | 11.41x |
|BM25 Top-X | 53.7 | 52.9 | 14.0 | 37.5 | 1.21x |
|SBERT Top-X | 39.5 | 38.8 | 7.3 | 37.5 | 1.21x |
|Dragon Top-X | 55.1 | 54.2 | 13.6 | 37.5 | 1.21x |
|RAPTOR Top-X | 52.0 | 51.2 | 11.8 | 35.1 | 1.13x |
|GARLIC | 61.1 | 60.2 | 18.6 | 31.0 | 1.00x |

| Qasper | F1 | ROUGE-L | BLEU-4 | TFLOPs | Ratio |
|---|---|---|---|---|---|
|Llama3.1 | 49.4 | 47.6 | 26.9 | 92.5 | 1.38x |
|MeMWalker | 39.0 | 36.8 | 17.4 | 123.9 | 1.85x |
|BM25 Top-X | 47.0 | 45.1 | 22.8 | 69.3 | 1.04x |
|SBERT Top-X | 46.6 | 44.5 | 23.3 | 68.9 | 1.03x |
|Dragon Top-X | 46.9 | 44.8 | 22.1 | 67.0 | 1.00x |
|RAPTOR Top-X | 46.9 | 44.7 | 20.8 | 67.3 | 1.01x |
|GARLIC | 49.7 | 47.9 | 27.0 | 66.9 | 1.00x |

| HotpotQA | F1 | ROUGE-L | BLEU-4 | TFLOPs | Ratio |
|---|---|---|---|---|---|
|Llama3.1 | 41.3 | 41.2 | 6.3 | 23.7 | 1.48x |
|MeMWalker | 39.7 | 38.9 | 13.9 | 93.4 | 5.84x |
|BM25 Top-X | 40.7 | 40.8 | 7.7 | 20.0 | 1.25x |
|SBERT Top-X | 40.8 | 40.7 | 7.5 | 19.6 | 1.23x |
|Dragon Top-X | 39.2 | 39.1 | 6.7 | 20.6 | 1.29x |
|RAPTOR Top-X | 40.7 | 40.7 | 7.2 | 17.9 | 1.12x |
|GARLIC | 43.5 | 43.5 | 7.2 | 16.0 | 1.00x |

| MuSiQue | F1 | ROUGE-L | BLEU-4 | TFLOPs | Ratio |
|---|---|---|---|---|---|
|Llama3.1 | 35.8 | 35.7 | 5.6 | 40.6 | 1.31x |
|MeMWalker | 24.0 | 23.5 | 9.9 | 175.7 | 5.69x |
|BM25 Top-X | 31.8 | 31.7 | 5.6 | 35.6 | 1.15x |
|SBERT Top-X | 32.5 | 32.5 | 6.4 | 35.6 | 1.15x |
|Dragon Top-X | 30.2 | 30.1 | 6.0 | 38.0 | 1.23x |
|RAPTOR Top-X | 35.4 | 35.2 | 7.2 | 32.2 | 1.04x |
|GARLIC | 36.9 | 36.8 | 5.7 | 30.9 | 1.00x |

After reordering Table 1 to list all Top-X results, it becomes clearer that our method outperforms all baselines with similar or even lower computational costs. For summary-based methods, MeMWalker, RAPTOR, and our approach, the computational costs for summarization are similar. However, our method still achieves superior performance compared to other summary-based methods at similar computational costs.

Our method introduces Dynamic Progress Control and a new attention-based retrieval paradigm leveraging Graph Search that does not rely on embeddings. Even with the same average TFLOPs, Dynamic Progress Control allows our method to better distribute and allocate computational resources across different queries and documents. Furthermore, as shown in Figure 4, our method allows stable adjustment of the trade-off between effectiveness and efficiency. These innovations are fundamentally different from existing RAG methods and are validated by our experimental results.

Among all baselines, our method is the only RAG approach that outperforms the LLM itself. As LLMs grow stronger and support longer input lengths, many RAG methods struggle to match the performance of LLMs, resulting in a widening performance gap. By deeply integrating the LLM into the retrieval process, our method achieves superior results, outperforming all baselines, including the LLM.

When compared to Llama 3.1, our method also mitigates scaling issues. For example, for a single-query document from NarrativeQA, Llama 3.1 requires 3361.9 TFLOPs, whereas our method requires only 2073.8 TFLOPs, encompassing both the summary and inference stages. Additionally, for multi-query documents, our method is significantly more efficient than Llama 3.1, as the summary graph can be reused across queries.

---

### Meta-Review · Area_Chair_KDsb · 2024-12-18

**Metareview:**

**Summary:**

This paper introduces GARLIC, a retrieval method for long document QA that effectively balances computational efficiency and accuracy. The core innovation lies in constructing a Hierarchical Weighted Directed Acyclic Graph (HWDAG), where text is summarized into fine-grained Information Points (IPs), each capturing single or few events. The graph edges are derived from LLM attention weights, enabling dynamic and flexible traversal across hierarchical levels.

Key contributions include:

* Dynamic Progress Control, which allows retrieval to stop when sufficient information is gathered.
* Attention-Based Retrieval, leveraging attention weights for relevance instead of embedding similarity.

**Strength:**

* Recursive use of a large language model to summarize long texts into information structures and assign attention-based edge weights introduces a novel information representation.
* GARLIC combines LLM-guided retrieval and computational efficiency, offering a valuable reference for other researchers.
* The method shows decent performance compared to existing retrieval approaches and outperforms Llama 3.1 while retaining RAG-like computational efficiency.

**Weakness:**

* The major weakness all reviewers pointed out is the lack of clarity and detailed explanation of the key components (e.g., edge weights by reviewer kpAv and LAnf; attention normalization by cbqx)
* Graph construction is resource-intensive since it needs multiple rounds of summarization using LLMs
* Lack of comparative analysis with other methods
* Further analysis is needed to explain why early termination in GARLIC performs better.
* Although embedding similarity improves performance, it introduces redundancy when compared to a purely attention-based retrieval method.

**Additional Comments On Reviewer Discussion:**

The authors provide detailed responses to the reviewers' comments and have revised the manuscript accordingly. However, I agree that the paper still lacks clarity and detailed explanations in several key issues raised by the reviewers. While some revisions were made during the rebuttal period, it remains unclear whether all concerns regarding clarity and explanation have been adequately addressed. Additionally, given that two reviewers hold a negative stance on the paper, rejection appears to be the appropriate decision.

---

### Decision · Program_Chairs · 2025-01-22

Reject